# A calibrated optogenetic toolbox of stable zebrafish opsin lines

Paride Antinucci[1][†], Adna Dumitrescu[2][†], Charlotte Deleuze[2], Holly J Morley[1], Kristie Leung[1], Tom Hagley[1], Fumi Kubo[3,4], Herwig Baier[4], Isaac H Bianco[1‡]*, Claire Wyart[2‡]*

[1]Department of Neuroscience, Physiology & Pharmacology, UCL, London, United Kingdom; [2]Institut du Cerveau et de la Moelle épinière (ICM), Sorbonne Universités, UPMC Univ Paris 06, Inserm, CNRS, Hôpital Pitié-Salpêtrière, Paris, France; [3]Center for Frontier Research, National Insitute of Genetics, Mishima, Japan; [4]Department Genes – Circuits – Behavior, Max Planck Institute of Neurobiology, Martinsried, Germany

**Abstract** Optogenetic actuators with diverse spectral tuning, ion selectivity and kinetics are constantly being engineered providing powerful tools for controlling neural activity with subcellular resolution and millisecond precision. Achieving reliable and interpretable in vivo optogenetic manipulations requires reproducible actuator expression and calibration of photocurrents in target neurons. Here, we developed nine transgenic zebrafish lines for stable opsin expression and calibrated their efficacy in vivo. We first used high-throughput behavioural assays to compare opsin ability to elicit or silence neural activity. Next, we performed in vivo whole-cell electrophysiological recordings to quantify the amplitude and kinetics of photocurrents and test opsin ability to precisely control spiking. We observed substantial variation in efficacy, associated with differences in both opsin expression level and photocurrent characteristics, and identified conditions for optimal use of the most efficient opsins. Overall, our calibrated optogenetic toolkit will facilitate the design of controlled optogenetic circuit manipulations.

*For correspondence:
i.bianco@ucl.ac.uk (IHB);
claire.wyart@icm-institute.org
(CW)

[†]These authors contributed
equally to this work
[‡]These authors also contributed
equally to this work

Competing interest: See
page 26

Reviewing editor: Harold
Burgess,

## Introduction

Optogenetics has greatly advanced our ability to investigate how neural circuits process information and generate behaviour by allowing manipulation of neural activity with high spatio-temporal resolution in genetically-defined neurons (*Miesenböck, 2009*; *Boyden, 2011*; *Miesenböck, 2011*; *Adamantidis et al., 2015*; *Boyden, 2015*; *Deisseroth, 2015*; *Deisseroth and Hegemann, 2017*). The efficacy with which optogenetic actuators – such as microbial opsins – can control neuronal spiking in vivo depends on biophysical properties, expression level and membrane trafficking of the opsin, physiological properties of the target cell and the intensity profile of light delivered within scattering tissue.

Accordingly, two primary experimental requirements should be met to enable controlled and reproducible in vivo optogenetic circuit manipulations: (*i*) reproducible opsin expression levels (across cells and animals), with stable expression systems offering higher reliability and homogeneity than transient ones (*Kikuta and Kawakami, 2009*; *Yizhar et al., 2011*; *Sjulson et al., 2016*), and (*ii*) calibrated photocurrents and effects on spiking recorded in target neurons (*Huber et al., 2008*; *Mardinly et al., 2018*; *Li et al., 2019*). While previous studies have compared the physiological effects of opsin activation in single cells using standardised conditions (e.g. *Berndt et al., 2011*; *Mattis et al., 2011*; *Prigge et al., 2012*; *Klapoetke et al., 2014*; *Berndt et al., 2016*; *Mardinly et al., 2018*), these comparisons were primarily performed in vitro or ex vivo using transient expression strategies.

In this study, we took advantage of the genetic accessibility and transparency of zebrafish (*Arrenberg et al., 2009*; *Del Bene and Wyart, 2012*; *Arrenberg and Driever, 2013*; *Portugues et al., 2013*; *Förster et al., 2017*) to generate nine stable transgenic lines for targeted opsin expression using the GAL4/UAS binary expression system (*Scheer and Campos-Ortega, 1999*; *Asakawa and Kawakami, 2008*) and quantitatively compare their efficacy for inducing or silencing neuronal spiking. We selected opsins that were reported to induce photocurrents with large amplitude (CoChR [*Klapoetke et al., 2014*], CheRiff [*Hochbaum et al., 2014*], ChR2$_{(H134R)}$[*Gradinaru et al., 2007*], eArch3.0 [*Mattis et al., 2011*], GtACR1,2 [*Govorunova et al., 2015*]) and/or fast kinetics (Chronos, ChrimsonR [*Klapoetke et al., 2014*], eNpHR3.0 [*Gradinaru et al., 2010*]). We first assessed the efficacy of these stable lines to control activity in intact neural populations via high-throughput behavioural assays at both embryonic and larval stages. Next, we made in vivo electrophysiological recordings from single low input-resistance motor neurons to calibrate photocurrents and test the ability of each line to elicit or silence spiking. We observed broad variation in behavioural response rates, photocurrent amplitudes and spike induction, likely due to differences in both opsin properties and expression levels. For the best opsin lines, we identified conditions that allowed control of individual action potentials within high-frequency spike trains. Overall, our toolkit will enable reliable and robust optogenetic interrogation of neural circuit function in zebrafish.

## Results

### Generation of stable transgenic lines for targeted opsin expression in zebrafish

To maximise the utility of our optogenetic toolkit, we used the GAL4/UAS binary expression system for targeted opsin expression in specific cell populations (*Figure 1*). We generated nine stable UAS lines for opsins having different ion selectivities and spectral tuning, fused to a fluorescent protein reporter (tdTomato or eYFP; *Figure 1A* and *Supplementary file 1*; *Asakawa et al., 2008*; *Arrenberg et al., 2009*; *Horstick et al., 2015*). GAL4 lines were used to drive expression in defined neuronal populations, such as motor neurons (*Figure 1B*; *Scott et al., 2007*; *Wyart et al., 2009*; *Böhm et al., 2016*). High levels of expression were achieved in most cases (*Figure 1C* and *Figure 1—figure supplement 1*), with only few opsins showing intracellular puncta suggestive of incomplete trafficking to the plasma membrane (CheRiff and GtACR2) or low expression (Chronos). To quantitatively compare opsin lines, we performed standardised behavioural tests at embryonic and larval stages (*Figure 1D*) and calibrated photocurrents and modulation of spiking in larval primary motor neurons (*Figure 1E*).

### Escape behaviour triggered by optogenetic activation of embryonic trigeminal neurons

As a first test of our opsin lines, we evaluated their ability to activate embryonic neurons (*Figure 2A–C*), which are characterised by high input resistance (*Drapeau et al., 1999*; *Saint-Amant and Drapeau, 2000*). We used the *Tg(isl2b:GAL4)* transgene (*Ben Fredj et al., 2010*) to drive expression of opsins in the trigeminal ganglion (*Figure 2B,C*). In this class of somatosensory neuron, optogenetic induction of few spikes has been shown to reliably elicits escape responses (*Douglass et al., 2008*), characterised by high-amplitude bends of the trunk and tail (*Kimmel et al., 1990*; *Saint-Amant and Drapeau, 1998*; *Sagasti et al., 2005*). Brief pulses of light (5 or 40 ms) induced escape responses in embryos (28–30 hr post fertilisation, hpf) expressing all cation- and anion-conducting channelrhodopsins (*Figure 2C–E* and *Figure 2—video 1*), while no movement was elicited in opsin-negative siblings (*Figure 2F,G* and *Figure 2—figure supplements 1* and *2*; N = 69 ± 26 fish per group, mean ± SD). The excitatory effect of GtACRs suggests that increasing chloride conductance depolarises neurons at this developmental stage. For all opsins, response probability increased monotonically with light power (*Figure 2F,G*). Escape behaviour could also be evoked via transient opsin expression, in which animals were tested one day after injection of DNA constructs into single cell-stage *Tg(isl2b:GAL4)* embryos (*Figure 2F*). Some opsins showed higher response probability in transient transgenic animals (CheRiff, CoChR and GtACRs), likely due to higher expression levels.

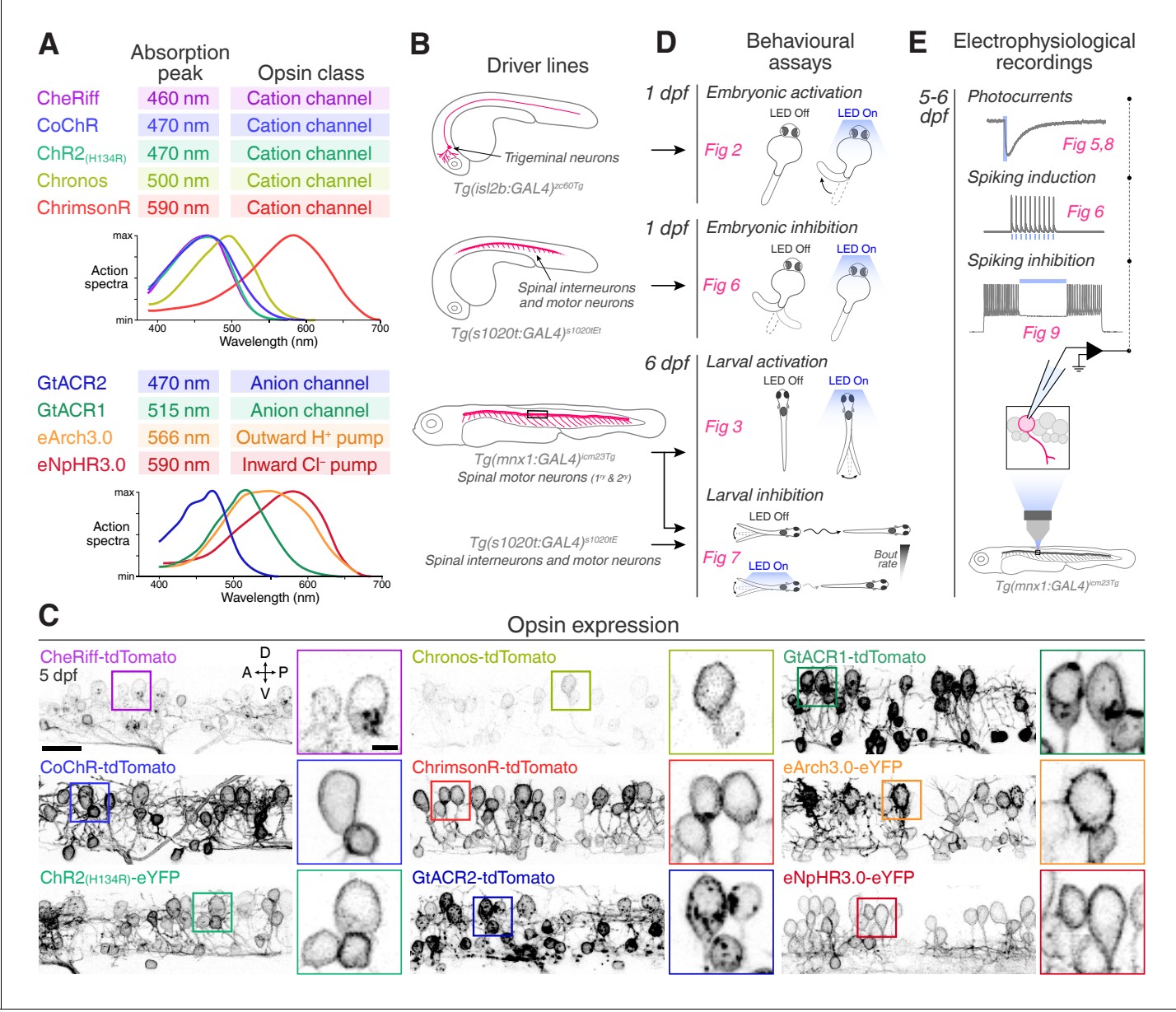

**Figure 1.** Toolkit for targeted opsin expression. (A) List of selected opsins, with spectral absorption and opsin class. (B) Schematics of expression patterns in the GAL4 transgenic driver lines used in this study. (C) Opsin expression in spinal neurons in *Tg(mnx1:GAL4;UAS:opsin-FP)* larvae at 5 dpf (for eNpHR3.0, the *s1020t:GAL4* transgene was used). Insets show magnified cell bodies to illustrate opsin membrane expression (for insets, brightness and contrast were adjusted independently for each opsin to aid visualisation). A, anterior; D, dorsal; P, posterior; V, ventral. Scale bar 20 $\mu$m in large images, 5 $\mu$m in insets. (D) Behavioural assays and corresponding figure numbers. (E) In vivo electrophysiological recordings and figure numbers. See also *Figure 1—figure supplement 1*.

The online version of this article includes the following figure supplement(s) for figure 1:

**Figure supplement 1.** Analysis of opsin expression in larval motor neurons.

With blue light, CoChR elicited escapes at the highest response probability (65–100% at 112–445 $\mu$W/mm$^2$; *Figure 2F,G*) and response latency decreased with increasing irradiance (insets in *Figure 2F,G*). As expected from its red-shifted absorption spectrum, ChrimsonR was the only cation channelrhodopsin to evoke escapes using amber light (~70% response probability at 322 $\mu$W/mm$^2$; *Figure 2F,G*; *Klapoetke et al., 2014*). Consistent with their respective red- and blue-shifted absorption spectra, GtACR1 triggered escapes upon amber and blue light stimulation whereas GtACR2 elicited responses only with blue light (*Figure 2F,G*; *Govorunova et al., 2015*).

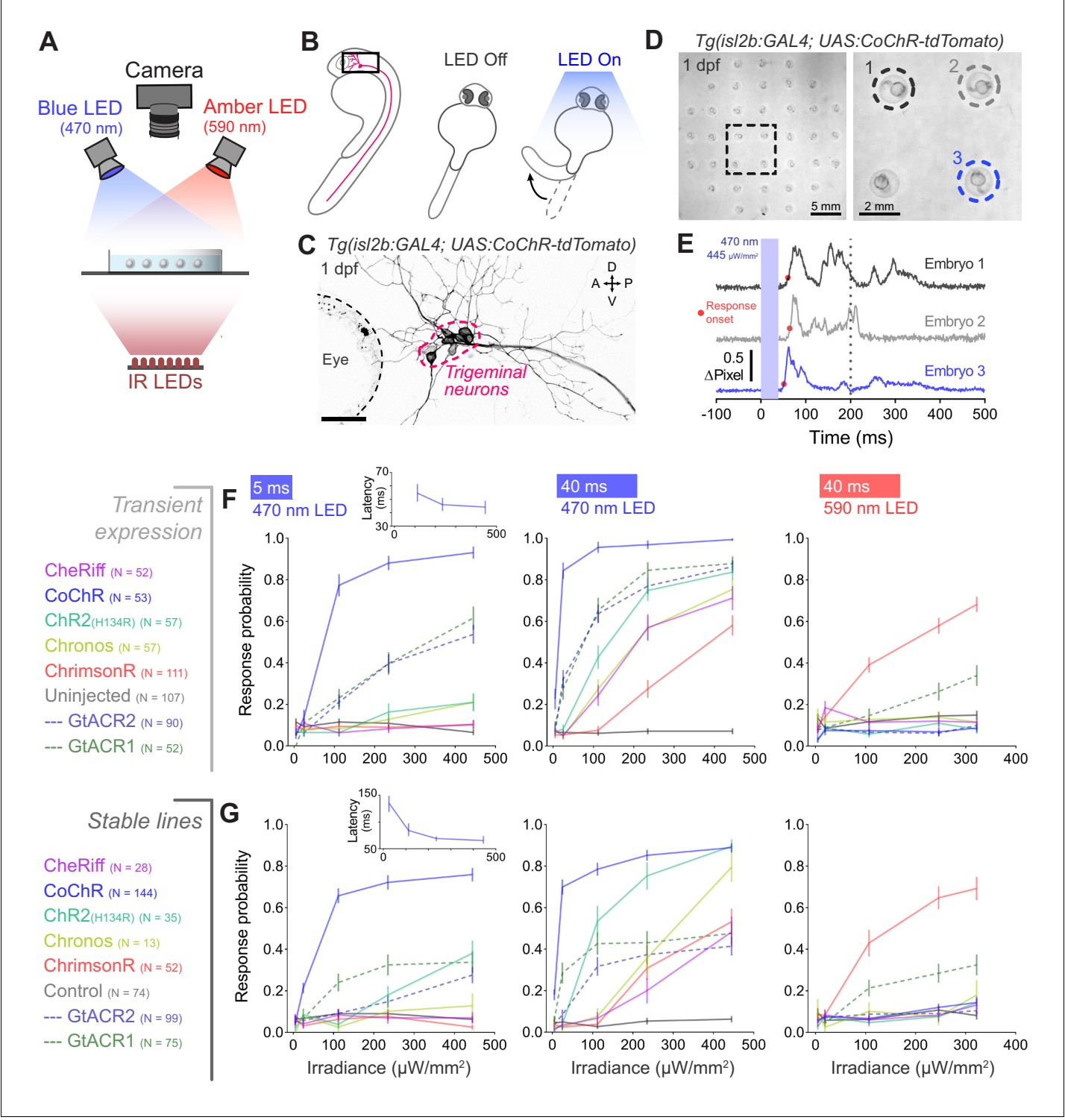

**Figure 2.** Optogenetic activation of embryonic trigeminal neurons triggers escape responses. (**A**) Experimental setup for optogenetic stimulation and behavioural monitoring. IR, infrared. (**B**) Schematic of behavioural assay. (**C**) Opsin expression in trigeminal neurons in a *Tg(isl2b:GAL4;UAS:CoChR-tdTomato)* embryo at 1 dpf. Imaging field of view corresponds to black box in (**B**). A, anterior; D, dorsal; P, posterior; V, ventral. Scale bar 50 $\mu$m. (**D**) *Tg(isl2b:GAL4;UAS:CoChR-tdTomato)* embryos positioned in individual agarose wells. Behaviour was monitored at 1000 frames per second across multiple embryos (28–30 hpf; N = 69 ± 26 fish per opsin group, mean ± SD) subjected to 5 or 40 ms pulses of full-field illumination (470 or 590 nm, 4.5–445 $\mu$W/mm$^2$) with a 15 s inter-stimulus interval. (**E**) Optogenetically-triggered escape responses detected from $\Delta$Pixel traces in the three embryos indicated in (**D**). Dotted line indicates maximum latency (200 ms) for a response to be considered optogenetically-triggered. (**F,G**) Response probability

*Figure 2 continued on next page*

*Figure 2 continued*

for transient (E) or stable (F) transgenic embryos expressing different opsins (mean ± SEM, across fish). Insets show response latency for 5 ms blue light pulses in CoChR-expressing embryos (median ± 95% CI, across fish). See also *Figure 2—figure supplements 1* and *2* and *Figure 2—video 1*.

The online version of this article includes the following video, source data, and figure supplement(s) for figure 2:

**Source data 1.** Data related to *Figure 2*.
**Figure supplement 1.** Response probability vs. time in transient transgenic embryos expressing opsins in trigeminal neurons.
**Figure supplement 2.** Response probability vs. time in stable transgenic embryos expressing opsins in trigeminal neurons.
**Figure 2—video 1.** Escape responses elicited by optogenetic stimulation of embryonic trigeminal neurons.
https://elifesciences.org/articles/54937#fig2video1

## Tail movements triggered by optogenetic activation of larval spinal motor neurons

Next, we compared the efficacy of cation channelrhodopsin lines to induce behaviour by activation of larval motoneurons, from which we would later record photocurrents. We used the *Tg(mnx1: GAL4)* transgene (*Böhm et al., 2016*) to target expression to spinal motor neurons (*Figure 3A,B*) and subjected head-restrained zebrafish (6 days post fertilisation, dpf; N = 28 ± 8 fish per group, mean ± SD) to either single light pulses (2 or 10 ms) or pulse trains at 20 or 40 Hz (*Figure 3C,D* and *Figure 3—videos 1* and *2*) while monitoring tail movements.

Optogenetically-evoked tail movements were triggered with short latency following light onset (8.3 ± 6.9 ms, mean ± SD) in opsin-expressing larvae only, whereas visually-evoked swim bouts occurred at much longer latency (316 ± 141 ms, mean ± SD) in both opsin-expressing larvae and control siblings (*Figure 3E*). We restricted our analyses to optogenetically-evoked movements, initiated within 50 ms of stimulus onset (corresponding to a minimum of the probability density distribution of latency; dotted line in *Figure 3E*). Optogenetically-evoked tail movements comprised a sequence of left-right alternating half beats, thereby resembling natural swim bouts (*Figure 3C,D* and *Figure 3—videos 1* and *2*). Response probability increased with irradiance (*Figure 3F* and *Figure 3—figure supplement 1*) and CoChR again elicited tail movements with the highest probability and shortest latency in response to blue light (96–100% at 0.63–2.55 mW/mm$^2$; *Figure 3F,G*). Only the ChrimsonR line responded to red light (~78% response probability at 1 mW/mm$^2$; *Figure 3F*). Tail movements evoked by single light pulses typically had shorter duration and fewer cycles than visually-evoked swims (*Figure 3H–K*). However, longer movements (>100 ms, 4–5 cycles) were often observed in response to single light pulses (see response to 2 ms pulse in *Figure 3D* and *Figure 3—video 1*) indicating engagement of spinal central pattern generators. This may occur through recruitment of glutamatergic V2a interneurons connected to motor neurons via gap junctions (*Song et al., 2016*) and/or by proprioceptive feedback via cerebrospinal fluid-contacting neurons (*Wyart et al., 2009*; *Fidelin et al., 2015*; *Böhm et al., 2016*). Pulse train stimuli evoked swim bouts of longer duration, with swims in CoChR and ChrimsonR lines showing modest frequency-dependent modulation of cycle number (*Figure 3L–Q*).

## In vivo whole-cell recording of photocurrents in larval primary motor neurons

To calibrate photocurrents in vivo, we performed whole-cell voltage clamp recordings from single primary motor neurons (pMNs) in 5–6 dpf larvae (*Figure 4A*). Each opsin was stimulated with a wavelength close to its absorption peak (1–30 mW/mm$^2$; *Figure 4—figure supplement 1A*). We recorded over 138 neurons, including control cells from opsin-negative animals, from which 90 cells were selected following strict criteria for recording quality (see Materials and methods; N = 3–19 included cells per group; *Figure 4—figure supplement 1B*). Opsin-expressing pMNs displayed physiological properties, such as membrane resistance, resting membrane potential and cell capacitance, comparable to control opsin-negative cells (*Figure 4B,C* and *Figure 4—figure supplement 1C,D*). All cation channelrhodopsins induced inward currents upon light stimulation, which were not observed in opsin-negative pMNs (*Figure 4D*). Notably, CoChR and ChrimsonR generated the largest photocurrents (CoChR 475 ± 186 pA, mean ± SD, N = 8 cells, ChrimsonR 251 ± 73 pA, N = 7; *Figure 4E*), consistent with their higher expression level (*Figure 1—figure supplement 1D*) and efficacy in behavioural assays (*Figures 2* and *3*). We did not observe significant irradiance-dependent

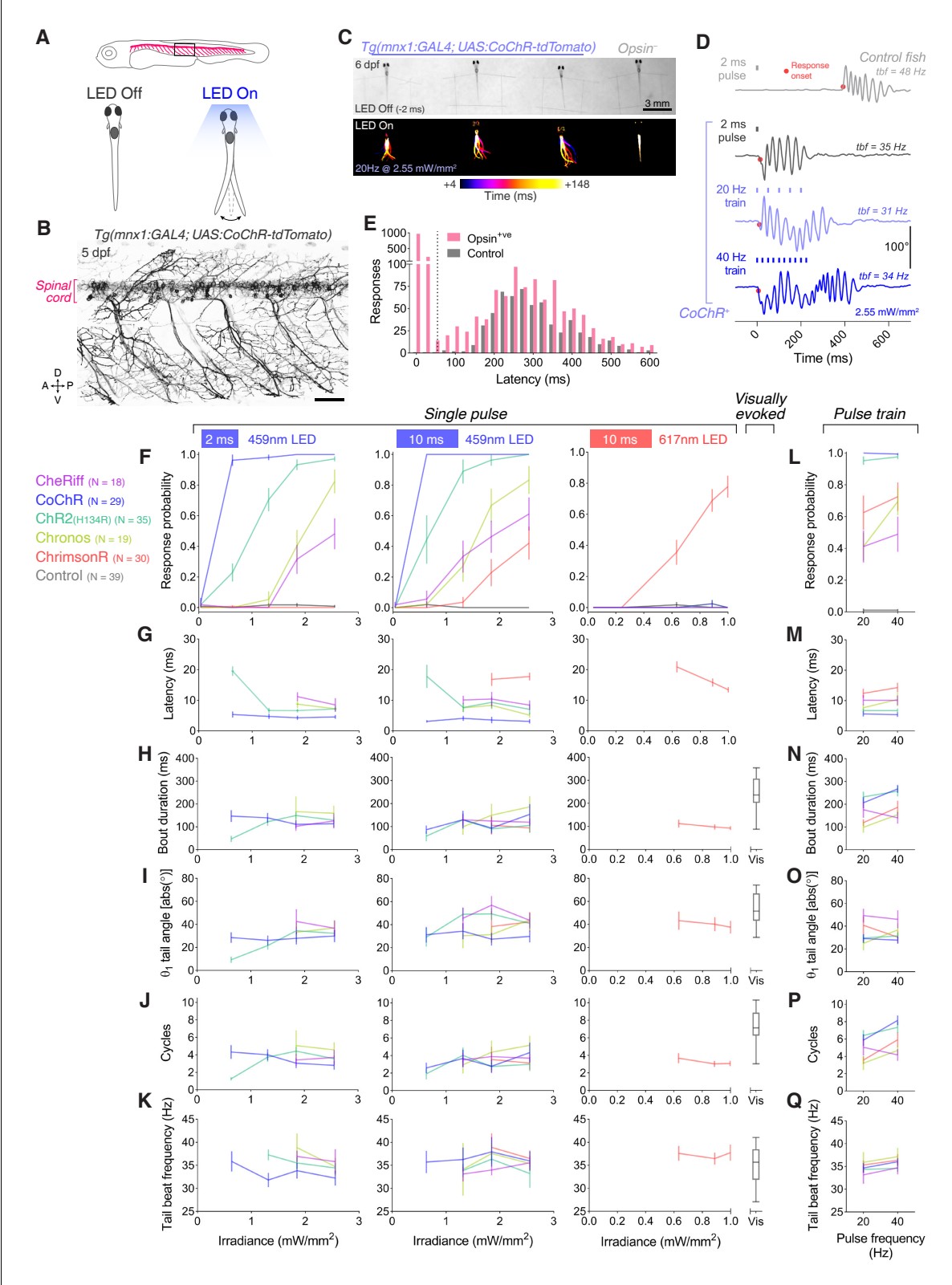

**Figure 3.** Optogenetic activation of larval spinal motor neurons triggers tail movements. (**A**) Schematics of behavioural assay. Head-restrained, tail-free larvae (6 dpf; N = 28 ± 8 fish per opsin group, mean ± SD) were exposed to 2 or 10 ms pulses of light (459 or 617 nm, 0.04–2.55 mW/mm$^2$) with a 20 s inter-stimulus interval while their behaviour was monitored at 500 fps. We also provided 250 ms trains of light pulses at 20 or 40 Hz. (**B**) Opsin expression in spinal motor neurons in a *Tg(mnx1:GAL4;UAS:CoChR-tdTomato)* larva at 5 dpf. Imaging field of view corresponds to black box in (**A**). A, *Figure 3 continued on next page*

*Figure 3 continued*

anterior; D, dorsal; P, posterior; V, ventral. Scale bar 50 $\mu$m. (C) Swim bouts elicited by a pulse train in *Tg(mnx1:GAL4;UAS:CoChR-tdTomato)* larvae (left). The control, opsin-negative larva (right), does not respond within 148 ms after stimulus onset. (D) Tail tracking, showing optogenetically-evoked swim bouts in a CoChR-expressing larva (bottom three rows) and a visually-evoked swim in a control opsin-negative larva (top). tbf, tail beat frequency. (E) Distribution of response latencies for all tail movements in opsin-expressing (red) and control opsin-negative larvae (grey). Dotted line indicates maximum latency (50 ms) for a response to be considered optogenetically-triggered. Control larvae exclusively show long latency responses. Each time bin corresponds to 25 ms. (F,L) Response probability of larvae expressing different opsins for single-pulse (F) or pulse-train (L) stimulation (mean ± SEM, across fish). G–Q Latency (G,M), bout duration (H,N), tail angle of the first half beat ($\theta_1$; I,O), number of cycles (J,P) and tail beat frequency (K,Q) for single-pulse (G–K) or pulse-train (M–Q) stimulation (mean ± SEM, across fish). See also *Figure 3—figure supplement 1* and *Figure 3—videos 1* and *2*. The online version of this article includes the following video, source data, and figure supplement(s) for figure 3:

**Source data 1.** Data related to *Figure 3*.

**Figure supplement 1.** Response probability vs. time in larvae expressing opsins in spinal motor neurons.

**Figure 3—video 1.** Swim bouts elicited by single-pulse optogenetic stimulation of larval spinal motor neurons.

https://elifesciences.org/articles/54937#fig3video1

**Figure 3—video 2.** Swim bouts elicited by 20 Hz pulse train optogenetic stimulation of larval spinal motor neurons.

https://elifesciences.org/articles/54937#fig3video2

modulation of photocurrent amplitude in any opsin line, likely due to the high range of irradiance we tested (*Figure 4—figure supplement 1F*). Photocurrent kinetics influence the temporal precision with which single action potentials can be evoked (*Mattis et al., 2011*). Therefore, we measured the photocurrent activation time (i.e. time to peak response from light onset), which results from the balance between activation and inactivation of the opsin, and deactivation time constant, which is determined by the rate of channel closure at light offset (*Mattis et al., 2011*; *Schneider et al., 2015*). Comparable activation times were observed across opsin lines (4–5 ms; *Figure 4F*). Deactivation time constants were more variable between opsins, with Chronos showing the fastest deactivation kinetics (4.3 ± 0.4 ms, N = 3 cells, mean ± SD) and the other opsins displaying longer time constants (12–20 ms; *Figure 4G*).

## Optogenetic induction of spiking in larval pMNs

To investigate whether our cation channelrhodopsin lines can induce action potentials in pMNs, we performed in vivo current clamp recordings while providing single light pulses (0.1–5 ms duration). In all opsin lines, light stimulation induced voltage depolarisations, which were never observed in opsin-negative pMNs, and voltage responses above –30 mV were classified as spikes (*Figure 5A*).

CoChR and ChrimsonR were the only opsin lines capable of triggering spiking in this cell type (*Figure 5A* and *Figure 5—figure supplement 1A–C*), as expected from their peak photocurrents exceeding pMN rheobase (dotted lines in *Figure 4E*). Notably, 5 ms light pulses induced spikes in all CoChR-expressing neurons (N = 11 out of 11 cells at 3–30 mW/mm$^2$), 92% of cells spiked with 1–2 ms pulses and only 50% spiked in response to 0.5 ms pulses (*Figure 5—figure supplement 1A*). ChrimsonR was less effective than CoChR in inducing action potentials, with 36–38% of neurons spiking when using 2–5 ms pulses (2 ms, N = 4 out of 11; 5 ms, N = 3 out of 8 cells) and only 1 cell out of 8 spiking in response to 1 ms pulses. In both lines, the number of evoked spikes increased with longer pulse duration (*Figure 5B* and *Figure 5—figure supplement 1D*).

For experiments aiming to replay physiological firing patterns, optogenetic actuators should be capable of inducing spike trains with millisecond precision and at biological firing frequencies. We thus tested the ability of CoChR and ChrimsonR to evoke pMN firing patterns across a range of frequencies (1–100 Hz; *Figure 5C*). Primary motor neurons can spike at high frequency (up to 300–500 Hz; *Menelaou and McLean, 2012*), hence optogenetic induction of high-frequency firing should not be limited by cell intrinsic physiological properties, but rather by opsin properties and light stimulation parameters. To assess the fidelity of firing patterns at each stimulation frequency, we measured spike number per light pulse as well as spike latency and jitter (i.e. standard deviation of spike latency). ChrimsonR could induce firing up to the highest frequency tested (100 Hz), with each light pulse typically evoking a single spike (*Figure 5C,D*). CoChR generated bursts of spikes in response to light pulses, even at the shortest stimulation duration and spiking consistently attenuated in the second half of the stimulation train (*Figure 5E,F*). Overall, spikes were induced with short latency (3–

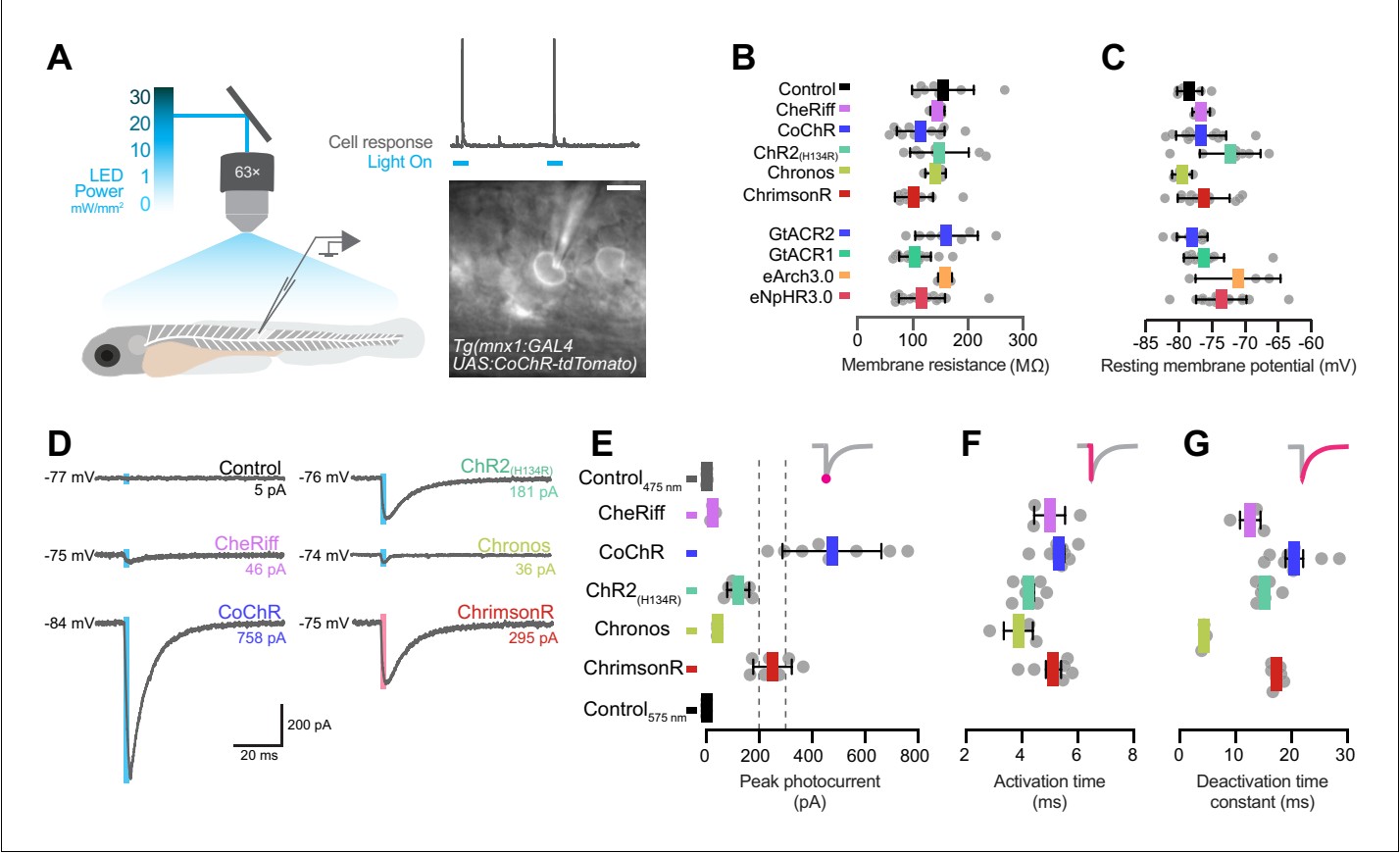

**Figure 4.** Electrophysiological recording of photocurrents in primary motor neurons. (**A**) Schematics of experimental setup for optogenetic stimulation with in vivo whole-cell patch clamp recordings. Image shows a patched primary motor neuron (pMN) expressing CoChR in a 6 dpf *Tg(mnx1:GAL4;UAS: CoChR-tdTomato)* larva. Scale bar 5 $\mu$m. (**B**) Membrane resistance was not affected by opsin expression (mean ± SD, across cells). (**C**) Resting membrane potential was similar between opsin-expressing and control neurons (mean ± SD). (**D**) Examples of inward photocurrents in response to 5 ms light pulses (20 mW/mm$^2$). (**E**) Peak photocurrent amplitude. CoChR and ChrimsonR induced the largest photocurrents (mean ± SEM, across cells). Dotted lines show range of pMN rheobase. Data is pooled across stimulus intensity (1–30 mW/mm$^2$) but see *Figure 4—figure supplement 1* for currents at varying irradiance. (**F**) Photocurrent activation time was similar across opsins (mean ± SEM). (**G**) Chronos photocurrents had the fastest deactivation time constant, while CoChR and ChrimsonR showed similar deactivation kinetics (mean ± SEM). See also *Figure 4—figure supplement 1*. The online version of this article includes the following source data and figure supplement(s) for figure 4:

**Source data 1.** Data related to *Figure 4*.
**Figure supplement 1.** Wavelengths used in electrophysiological recordings and photocurrent amplitude and kinetics as a function of irradiance.

4 ms mean latency) and low jitter (0.25–1.25 ms jitter) with both opsin lines (*Figure 5G,H* and *Figure 5—figure supplement 1E*).

## Optogenetic suppression of coiling behaviour in embryos

Next, we tested the ability of our opsin lines to suppress spontaneous behaviour of zebrafish embryos (*Saint-Amant and Drapeau, 1998*; *Warp et al., 2012*; *Mohamed et al., 2017*; *Bernal Sierra et al., 2018*). We targeted expression of the anion-conducting channels GtACR1 and GtACR2 (*Govorunova et al., 2015*), the outward proton pump eArch3.0 (*Mattis et al., 2011*) and the inward chloride pump eNpHR3.0 (*Gradinaru et al., 2010*) to spinal cord neurons using the *Tg (s1020t:GAL4)* transgene (*Scott et al., 2007*) and examined changes in spontaneous coiling behaviour in response to light (*Figure 6A–D* and *Figure 6—video 1*). Embryos were tested between 24 and 27 hpf, a stage at which embryos coil spontaneously (*Saint-Amant and Drapeau, 1998*) but show only minimal light-induced photomotor responses, which mostly occur later in development (30–40 hpf) (*Kokel et al., 2013*). In opsin-expressing embryos, light exposure led to a suppression of coiling behaviour that was followed by a synchronised restart at light offset (*Figure 6D,E* and

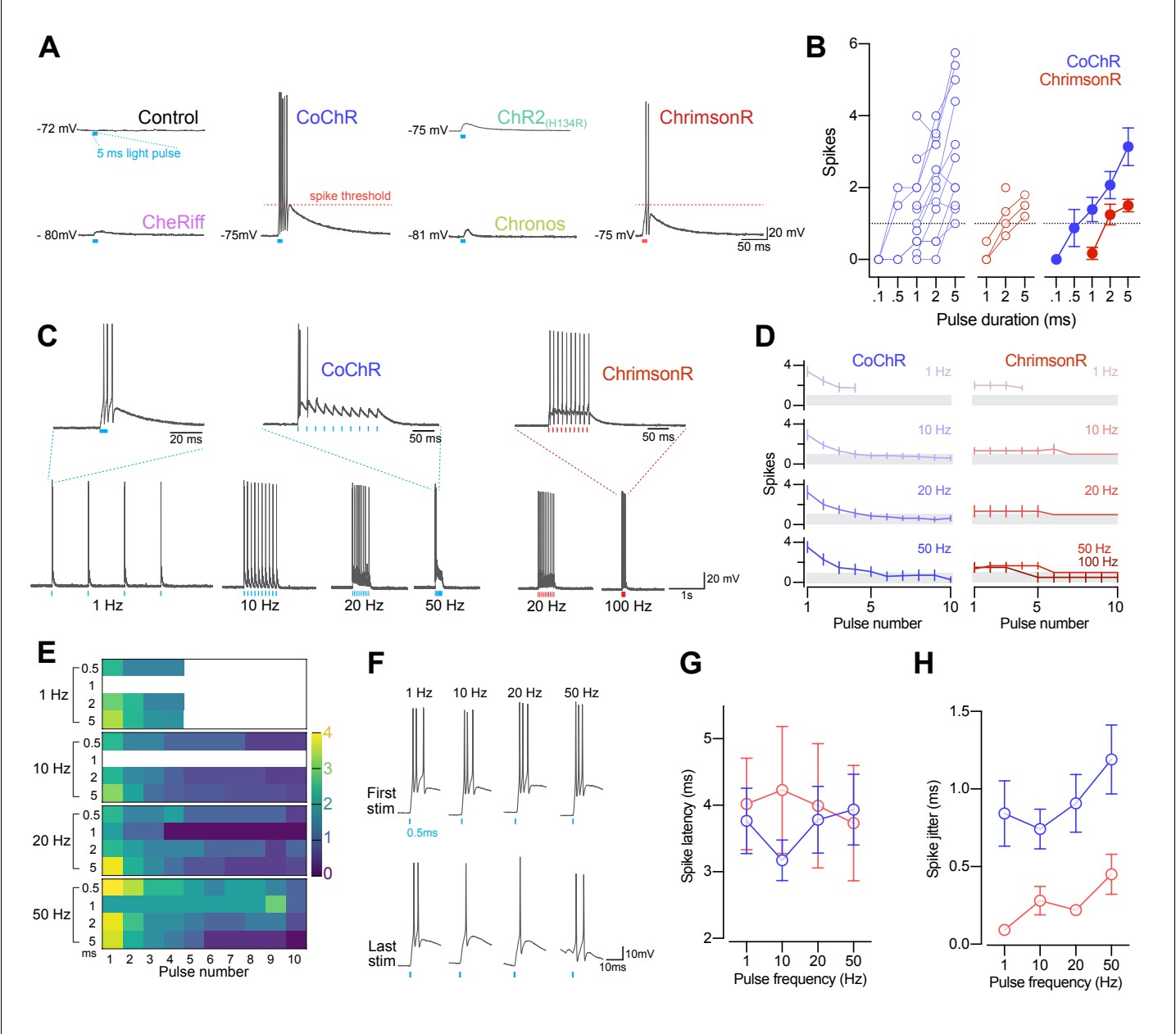

**Figure 5.** CoChR and ChrimsonR can elicit spiking in primary motor neurons. (**A**) Example membrane depolarisations induced by 5 ms light pulses (20 mW/mm$^2$). (**B**) Number of optogenetically-evoked spikes vs. pulse duration (across irradiance levels 1–30 mW/mm$^2$). Longer pulse duration induced more spikes in both CoChR- and ChrimsonR-expressing cells. Left plots show single neurons and right plot shows mean ± SEM across cells. (**C**) Example voltage responses from CoChR- and ChrimsonR-expressing cells upon pulse train stimulation (1–100 Hz, 2–5 ms pulse duration). (**D**) Number of spikes vs. pulse number within a train (mean ± SEM, across cells; shaded area depicts average number of spikes is below 1). In CoChR-expressing cells, the initial 3–4 pulses within the train induced bursts of 2–4 spikes. (**E**) Heatmap of mean spike number elicited via CoChR stimulation, separated according to stimulation frequency and pulse duration. Primary motor neurons often responded with bursts of action potentials, even for short light pulses. (**F**) Example responses to the 1st (top) and last (bottom) 0.5 ms light pulse in a train, recorded from a CoChR-positive neuron. (**G**) Spike latency vs. pulse frequency (mean ± SEM). (**H**) Spike jitter (mean ± SEM) vs. pulse frequency shows that ChrimsonR-expressing cells exhibited lower spike jitter than CoChR-expressing cells. See also *Figure 5—figure supplement 1*.

The online version of this article includes the following source data and figure supplement(s) for figure 5:

**Source data 1.** Data related to *Figure 5*.

**Figure supplement 1.** Optogenetically-evoked voltage responses as a function of irradiance and pulse frequency.

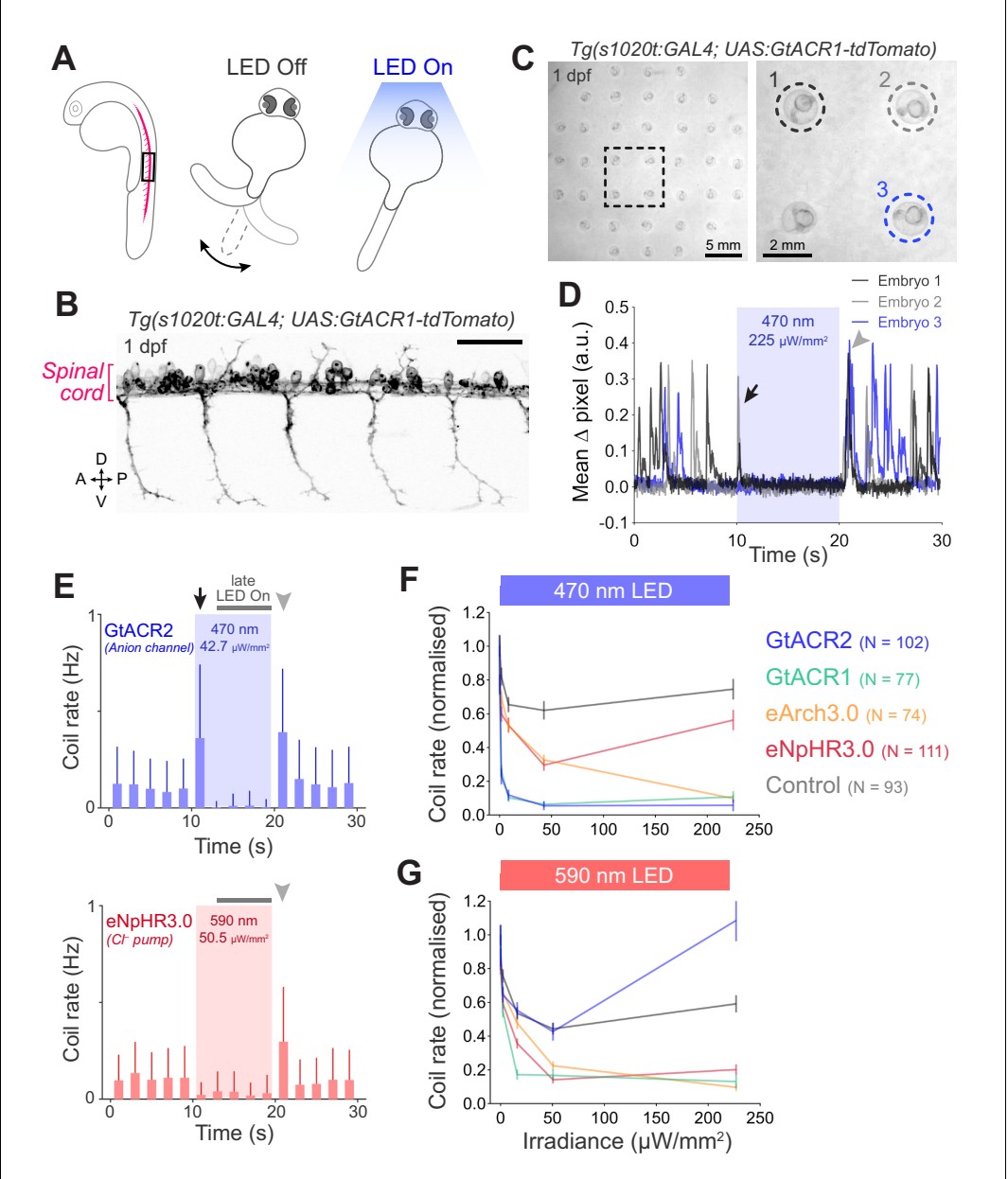

**Figure 6.** Optogenetic suppression of coiling behaviour in embryos. (A) Schematic of the behavioural assay. (B) Opsin expression in spinal motor neurons and interneurons in a *Tg(s1020t:GAL4;UAS:GtACR1-tdTomato)* embryo at 1 dpf. Imaging field of view corresponds to black box in (A). A, anterior; D, dorsal; P, posterior; V, ventral. Scale bar 50 $\mu$m. (C) Camera field of view showing *Tg(s1020t:GAL4;UAS:GtACR1-tdTomato)* embryos positioned in individual agarose wells. Behaviour was monitored at 50 frames per second across multiple embryos (24–27 hpf; N = 91 ± 16 fish per group, mean ± SD) subjected to 10 s light periods (470 or 590 nm, 0–227 $\mu$W/mm$^2$) with a 50 s inter-stimulus interval. (D) Tracking of coiling behaviour (mean $\Delta$Pixel from three trials) for the three embryos shown in (C). Black arrow indicates movements at light onset, whereas grey arrowhead indicates synchronised restart of coiling behaviour following light offset. (E) Optogenetically-induced changes in coil rate (mean + SD, across fish) in embryos expressing the anion channelrhodopsin GtACR1 (N = 77 embryos, top) or the Cl$^-$ pump eNpHR3.0 (N = 111 embryos, bottom). Horizontal dark grey bars indicate the' late LED On' period. Each time bin corresponds to 2 s. (F,G) Normalised coil rate during the' late LED On' period in embryos expressing different opsins (mean ± SEM, across fish). Control opsin-negative siblings were subjected to the same light stimuli. See also *Figure 6—figure supplements 1* and *2* and *Figure 6—video 1*.

The online version of this article includes the following video, source data, and figure supplement(s) for figure 6:

*Figure 6 continued on next page*

*Figure 6 continued*

**Source data 1.** Data related to *Figure 6*.
**Figure supplement 1.** Coil rate vs. time in embryos expressing different opsins in spinal neurons.
**Figure supplement 2.** Coil rate vs. irradiance for the initial 2 s of light exposure.
**Figure 6—video 1.** Monitoring of coiling behaviour upon opsin activation in embryonic spinal neurons.
https://elifesciences.org/articles/54937#fig6video1

---

*Figure 6—figure supplement 1*; N = 91 ± 16 fish per group, mean ± SD), as previously reported (*Warp et al., 2012*; *Mohamed et al., 2017*). As expected from behaviour with *Tg(isl2b:GAL4)* embryos (*Figure 2F,G*), GtACR activation in spinal neurons occasionally induced movements in the initial 1–2 s following light onset (black arrows in *Figure 6D,E*), a phenomenon that was not observed with Cl⁻/H⁺ pumps. Given these two effects, changes in coil rate were separately quantified for the initial 2 s (*Figure 6—figure supplement 2*) and subsequent 8 s period of light exposure ('late LED ON'; grey horizontal bars in *Figure 6E*).

All opsin lines suppressed coiling behaviour during the 'late LED ON' period (*Figure 6F,G*). As previously observed (*Friedmann et al., 2015*), light also decreased coiling in control opsin-negative embryos, yet to a significantly lesser degree than in opsin-expressing animals (*Figure 6F,G*). Optogenetically evoked suppression was likely a result of distinct mechanisms in the different transgenic lines. While Cl⁻/H⁺ pumps systematically induce hyperpolarisation, anion channelrhodopsins can silence cells via shunting as well as depolarisation block depending upon the reversal potential of chloride in vivo (see below and Discussion). GtACRs achieved the strongest suppression of coil rate using blue light (90–95% decrease at 8.4–225 $\mu$W/mm$^2$; *Figure 6F*). With amber light, GtACR1, eArch3.0 and eNpHR3.0 showed comparable suppression (80–90% decrease at 50.5–227 $\mu$W/mm$^2$), with GtACR1 achieving ~83% decrease in coil rate even at low irradiance (15.9 $\mu$W/mm$^2$; *Figure 6G*).

## Optogenetic suppression of swimming in larvae

To compare the efficacy of our opsin lines to suppress behaviour in larvae, we targeted opsin expression to spinal motor neurons and interneurons using *Tg(s1020t:GAL4)*, as above, and examined changes in spontaneous swimming behaviour of 6 dpf animals in response to 10 s light pulses (*Figure 7A–C* and *Figure 7—video 1*; N = 25 ± 9 fish per group, mean ± SD).

Expression of GtACR1, GtACR2 and eArch3.0 in motor neurons and interneurons reduced swim bout rate relative to control larvae in response to blue light, with GtACRs achieving the greatest suppression (20–45% decrease; *Figure 7D,E*; *Sternberg et al., 2016*). Consistent with a previous report (*Andalman et al., 2019*), opsin-negative larvae showed a 20–30% increase in bout rate during illumination with blue light (*Figure 7E* and *Figure 7—figure supplement 1*), while no increase was observed with red light (*Figure 7F*). Using red light, only eNpHR3.0 could reduce bout rate and suppression increased with higher irradiance (45% decrease at 1 mW/mm$^2$; *Figure 7F*). No increase in bout rate was found in larvae expressing anion channelrhodopsins even when analysis was restricted to the initial 2 s of the light period (*Figure 7—figure supplement 2A*), suggesting GtACRs do not induce excitatory effects at larval stages. Opsin activation did not affect bout speed (*Figure 7—figure supplement 2B*). By contrast, using the *Tg(mnx1:GAL4)* transgene to selectively drive expression only in motor neurons resulted in a decrease in bout speed (~20% reduction), but not bout rate (*Figure 7—figure supplements 3* and *4*).

## Photocurrents induced by anion channelrhodopsins and chloride/proton pumps

To analyse the physiological effects induced by anion channelrhodopsins and Cl⁻/H⁺ pumps, we measured their photocurrents through in vivo voltage clamp recordings from larval pMNs (5–6 dpf). Since anion channelrhodopsin function depends on chloride homeostasis (*Figure 8A*; *Govorunova et al., 2015*) and chloride reversal potential (ECl) is known to change over development (*Ben-Ari, 2002*; *Reynolds et al., 2008*; *Zhang et al., 2010*), we recorded GtACR1 photocurrents using two intracellular solutions: one mimicking ECl in embryonic neurons (–50 mV; *Saint-Amant and Drapeau, 2003*) and the second approximating intracellular chloride

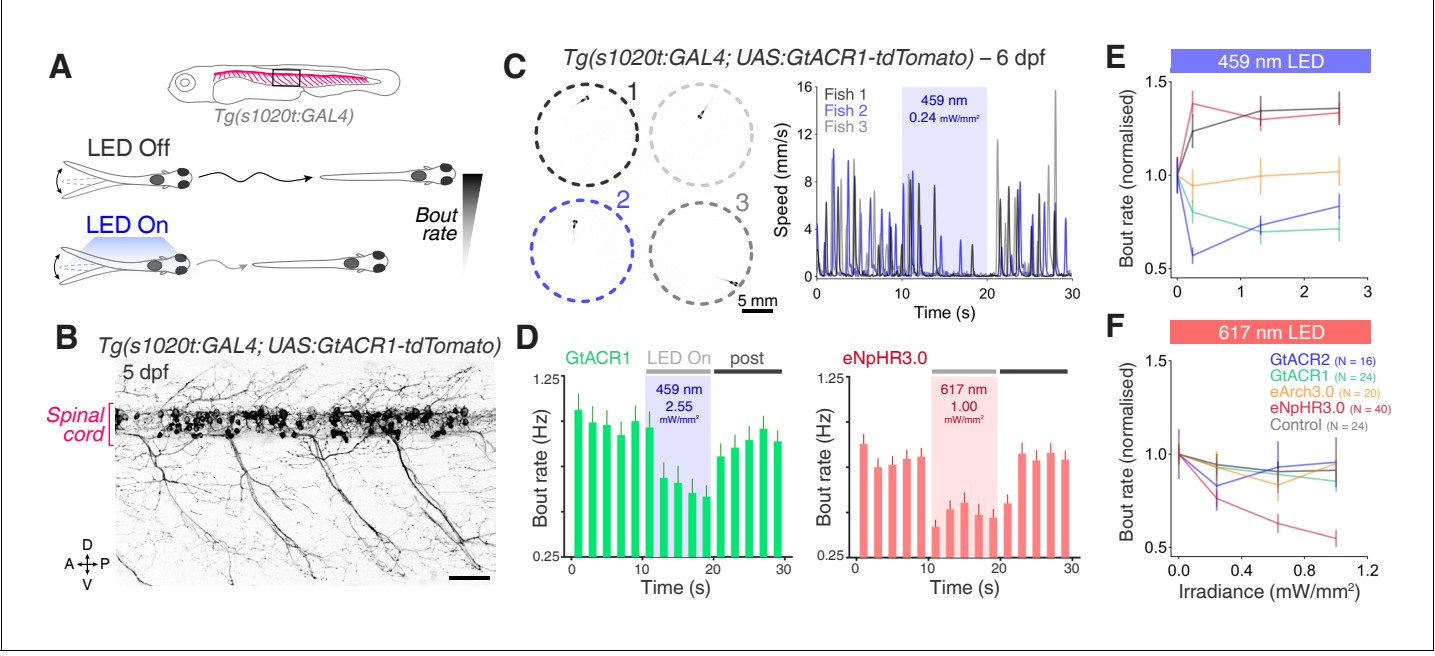

**Figure 7.** Optogenetic suppression of swimming in larvae. (**A**) Schematic of behavioural assay. (**B**) Opsin expression in spinal motor neurons and interneurons in a *Tg(s1020t:GAL4;UAS:GtACR1-tdTomato)* larva at 5 dpf. Imaging field of view corresponds to black box in (**A**). A, anterior; D, dorsal; P, posterior; V, ventral. Scale bar 50 $\mu$m. (**C**) *Tg(s1020t:GAL4;UAS:GtACR1-tdTomato)* larvae were positioned in individual agarose wells (left) and instantaneous swim speed was monitored by centroid tracking (right) at 50 fps (six dpf; N = 25 ± 9 fish per group, mean ± SD). 10 s light periods were delivered (459 or 617 nm, 0–2.55 mW/mm$^2$) with a 50 s inter-stimulus interval. (**D**) Optogenetically-induced changes in bout rate (mean + SEM, across fish) in *Tg(s1020t:GAL4)* larvae expressing GtACR1 (N = 24 larvae, left) or eNpHR3.0 (N = 40 larvae, right). Horizontal grey bars indicate the time windows used to quantify behavioural changes. Each time bin corresponds to 2 s. (**E,F**) Normalised bout rate during the 'LED On' period in larvae expressing different opsins (mean ± SEM, across fish) and in control, opsin-negative, siblings. See also *Figure 7—figure supplements 1–4* and *Figure 7—video 1*.

The online version of this article includes the following video, source data, and figure supplement(s) for figure 7:

**Source data 1.** Data related to *Figure 7*.
**Figure supplement 1.** Bout rate vs. time in larvae expressing different opsins in spinal neurons.
**Figure supplement 2.** Bout rate and speed vs. irradiance during different time periods in *Tg(s1020t:GAL4)* larvae.
**Figure supplement 3.** Optogenetic suppression of swimming in *Tg(mnx1:GAL4)* larvae.
**Figure supplement 4.** Bout rate and speed vs. irradiance during different time periods in *Tg(mnx1:GAL4)* larvae.
**Figure 7—video 1.** Suppression of swimming upon opsin activation in larval spinal neurons.
https://elifesciences.org/articles/54937#fig7video1

concentration in more mature, larval neurons (ECl = –70 mV, see Materials and methods). Inspection of I-V curves for GtACR1 photocurrents showed that, in both solutions, currents reversed with a positive 5–10 mV shift relative to ECl (*Figure 8—figure supplement 1A,B*), as previously observed (*Govorunova et al., 2015*) and within the expected error margin given our access resistance (*Figure 4—figure supplement 1C*; estimated voltage error for ECl$_{-50\ mV}$ solution, 4.6 ± 6.4 mV, mean ± SD, N = 5 cells; ECl$_{-70\ mV}$ solution, 1.2 ± 1.3 mV, N = 3). This suggests that GtACR1 photocurrents were primarily driven by chloride ions, as expected (*Govorunova et al., 2015*). The other opsin lines were tested using the ECl$_{-50\ mV}$ solution only. Neurons were stimulated with light (1 s pulse) at a holding potential matching their measured resting membrane potential (*Figure 4C*).

Anion channelrhodopsins induced inward, 'depolarising' photocurrents (as expected from the combination of ECl and holding potential), while Cl$^-$/H$^+$ pumps generated outward, 'hyperpolarising' currents (*Figure 8B*). All opsins except eNpHR3.0 showed bi-phasic photocurrent responses comprising a fast activation followed by a slow inactivation (*Figure 8B*), likely due to a fraction of the opsin population transitioning to an inactive state (*Chow et al., 2010*; *Mattis et al., 2011*; *Schneider et al., 2015*). We measured both the peak photocurrent (*Figure 8C*) as well as the steady-state current during the last 5 ms of the light period (*Figure 8D*). GtACRs induced

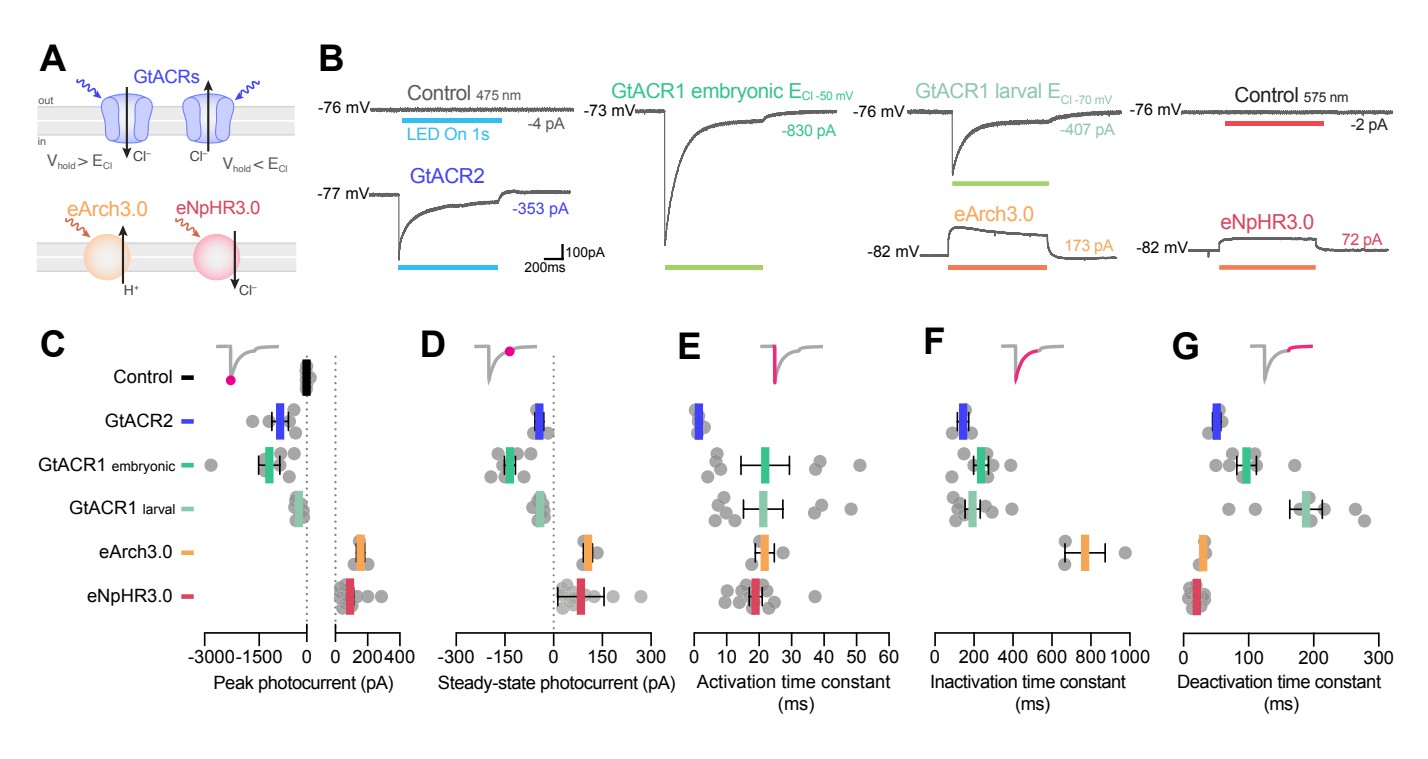

**Figure 8.** Photocurrents induced by anion channelrhodopsins and chloride/proton pumps. (A) Action of anion channelrhodopsins (top) and Cl⁻/H⁺ pumps (bottom). For anion channelrhodopsins, photocurrent magnitude and direction depend on chloride reversal potential (ECl) and holding potential (V$_{hold}$), while Cl⁻/H⁺ pumps always induce outward currents. (B) Example photocurrents in response to a 1 s light exposure (20 mW/mm²). (C, D) Photocurrent peak (C) and steady-state (D) amplitude (mean ± SEM, across cells). GtACRs induced larger photocurrents than Cl⁻/H⁺ pumps. (E–G) Photocurrent activation (E), inactivation (F) and deactivation (G) time constants (mean ± SEM). Photocurrents induced by Cl⁻/H⁺ pumps showed minimal inactivation and faster deactivation kinetics than GtACRs. eNpHR3.0 photocurrents did not inactivate hence no inactivation time constant was computed. See also *Figure 8—figure supplement 1*.

The online version of this article includes the following source data and figure supplement(s) for figure 8:

**Source data 1.** Data related to *Figure 8*.

**Figure supplement 1.** Photocurrent amplitude and kinetics as a function of irradiance.

photocurrents with peak amplitude 3–10 times larger than those generated by Cl⁻/H⁺ pumps (*Figure 8C*), while steady-state currents were similar across opsins (*Figure 8D*). Some degree of irradiance-dependent modulation of photocurrents was observed, primarily in peak amplitude (*Figure 8—figure supplement 1C–E*). To characterise photocurrent kinetics, we computed activation, inactivation and deactivation time constants (*Mattis et al., 2011*). GtACR photocurrents had the fastest activation kinetics (~1 ms at 30 mW/mm²; *Figure 8E* and *Figure 8—figure supplement 1F*). However, deactivation kinetics of Cl⁻/H⁺ pumps were 2–10 times faster than those induced by GtACRs (14–22 ms eNpHR3.0, 27–37 ms eArch3.0; *Figure 8G* and *Figure 8—figure supplement 1H*) and showed little inactivation (600–1000 ms eArch3.0; *Figure 8F* and *Figure 8—figure supplement 1G*).

## Optogenetic inhibition of pMN spiking

To investigate the ability of anion channelrhodopsins and Cl⁻/H⁺ pumps to suppress neural activity, we recorded pMNs in current clamp mode. In control opsin-negative neurons, light delivery (1 s) induced negligible voltage deflections (*Figure 9A*). By contrast, anion channelrhodopsins generated membrane depolarisation towards ECl while the Cl⁻/H⁺ pumps hyperpolarised the cell (*Figure 9A*), in accordance with recorded photocurrents. The absolute peak amplitude of voltage deflections was comparable between opsin lines (10–25 mV), with 10–40% decrease between peak and steady-state responses in all cases except eNpHR3.0, which generated stable hyperpolarisation (*Figure 9B,C* and

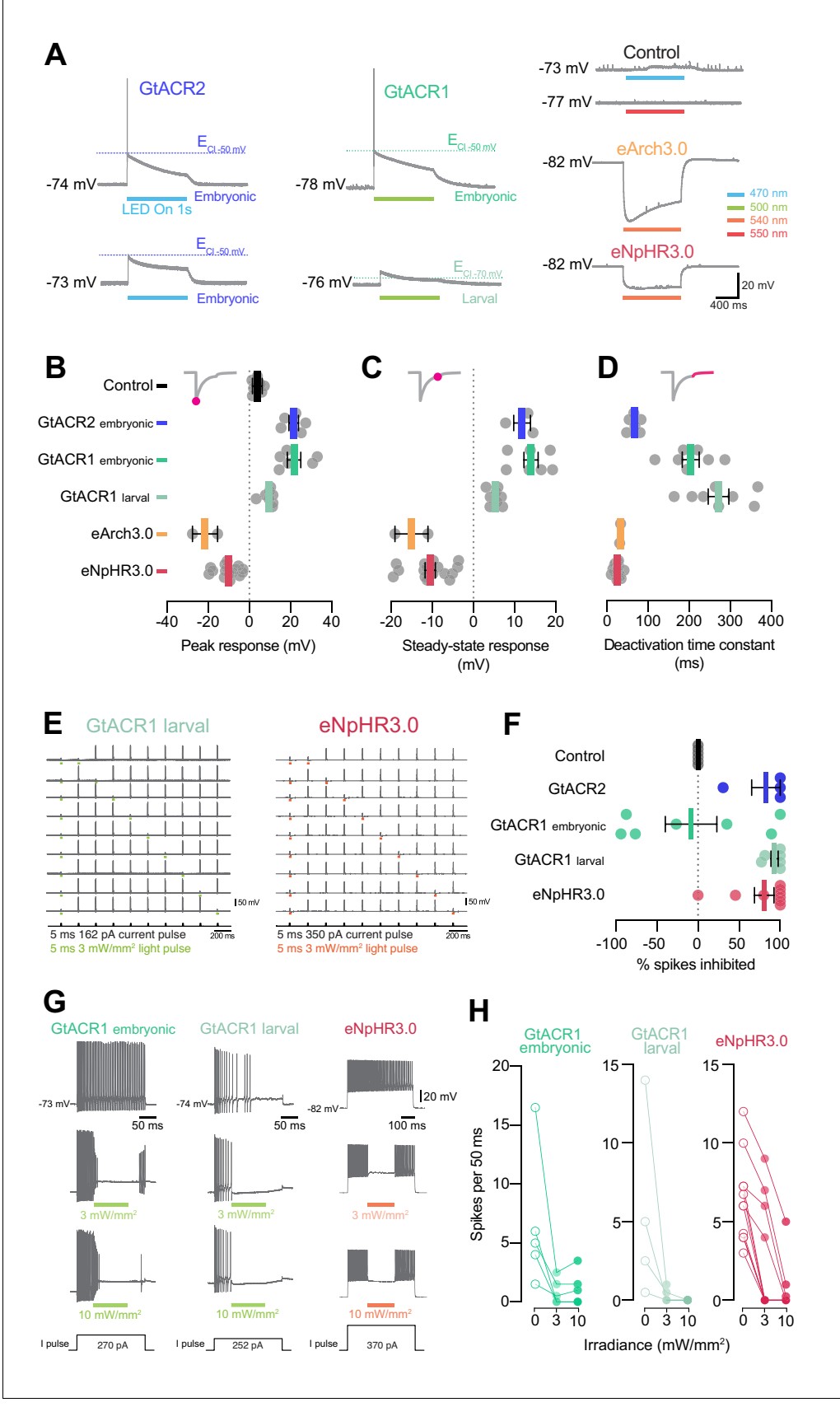

**Figure 9.** GtACRs and eNpHR3.0 effectively inhibited spiking. (**A**) Example voltage deflections induced by anion channelrhodopsins and Cl$^-$/H$^+$ pumps in response to a 1 s light pulse (20 mW/mm$^2$). (**B–D**) Peak (**B**) and steady-state (**C**) responses and deactivation time constant (**D**) of voltage deflections. All opsins induced similar absolute voltage changes. Anion channelrhodopsins generated depolarisation with both intracellular solutions while Cl$^-$/H$^+$ pumps generated hyperpolarisation. (**E**) Example recordings demonstrating inhibition of single spikes in GtACR1- and eNpHR3.0-expressing cells with 5 ms light pulses (3 mW/mm$^2$). (**F**) Fraction of spikes that were optogenetically inhibited (mean ± SEM, across cells). All opsins achieved high suppression efficacy, but GtACR1 induced additional spikes upon light delivery with the embryonic intracellular solution. (**G**) Example recordings demonstrating inhibition of sustained spiking in GtACR1- and eNpHR3.0-expressing cells. (**H**) Quantification of suppression using protocol illustrated in (**G**). Number of spikes per 50 ms during light delivery (0–10 mW/mm$^2$) is plotted against irradiance. GtACR1 and eNpHR3.0 inhibited tonic spiking with similar efficacy (mean ± SEM). See also *Figure 9—figure supplement 1*.

The online version of this article includes the following source data and figure supplement(s) for figure 9:

**Source data 1.** Data related to *Figure 9*.

**Figure supplement 1.** Optogenetically-evoked voltage responses as a function of irradiance.

---

*Figure 9—figure supplement 1A,B*). In a subset of GtACR1- (N = 4 out of 7) and GtACR2-expressing neurons (N = 2 out of 6), spiking was induced at light onset when using the ECl$_{-50 mV}$ solution (*Figure 9A*; GtACR1 6.7 ± 7.1 spikes; GtACR2 1.5 ± 0.7, mean ± SD). This is consistent with the movements evoked at light onset in young, 1 dpf embryos expressing GtACRs (*Figures 2* and *6*). The kinetics of voltage decay to baseline following light offset matched those of recorded photocurrents (*Figure 9D* and *Figure 9—figure supplement 1C*).

Next, we compared the utility of our opsin lines to inhibit pMN firing. First, we induced larval pMNs to fire at 5 Hz by injecting pulses of depolarising current (5 ms, 1.2–1.5 × rheobase) and simultaneously delivered 5 ms light pulses to inhibit selected spikes (*Figure 9E*). We found that GtACRs and eNpHR3.0 could effectively inhibit spikes (80–95% suppression), while light pulses did not alter firing in opsin-negative neurons (*Figure 9F*). In agreement with our current clamp recordings, a subset of GtACR1-expressing neurons (N = 4 out of 7) tested in the embryonic ECl$_{-50 mV}$ solution failed to suppress spikes and instead induced extra action potentials in response to light pulses, resulting in a negative spike inhibition efficacy (*Figure 9F*). Data from eArch3.0-expressing neurons could not be collected due to degradation in the quality of recordings or cells becoming highly depolarised (i.e. resting membrane potential > –50 mV) by the later stages of the protocol, suggesting that repeated eArch3.0 activation may alter electrical properties of neurons (*Williams et al., 2019*).

Lastly, we asked whether we could inhibit firing over periods of tens to hundreds of milliseconds. We injected long pulses of depolarising current (200–800 ms) to elicit tonic pMN firing, and simultaneously provided shorter light pulses (50–200 ms; 3–10 mW/mm$^2$) in the middle of the spike train (*Figure 9G*). Both GtACR1 and eNpHR3.0 successfully inhibited spiking during the light pulse, with complete suppression in 60–100% of cells at 10 mW/mm$^2$ irradiance (*Figure 9G,H*). Notably, GtACR1 could inhibit tonic spiking even when using the embryonic ECl$_{-50 mV}$ solution (*Figure 9G,H*), consistent with the suppression of coiling behaviour upon prolonged illumination of GtACR-expressing embryos (*Figure 6*).

## Discussion

In this study, we generated a set of stable transgenic lines for GAL4/UAS-mediated opsin expression in zebrafish and evaluated their efficacy in controlling neural activity in vivo. High-throughput behavioural assays and whole-cell electrophysiological recordings provided complementary insights to guide tool selection (*Figure 10*). Behavioural assays enabled efficient evaluation of opsin lines in various sensory and motor cell types and revealed developmental stage-specific effects in intact neural populations. Electrophysiological recordings from single motor neurons afforded quantification of photocurrents and systematic evaluation of the ability of these optogenetic tools to elicit or silence activity at single action potential resolution.

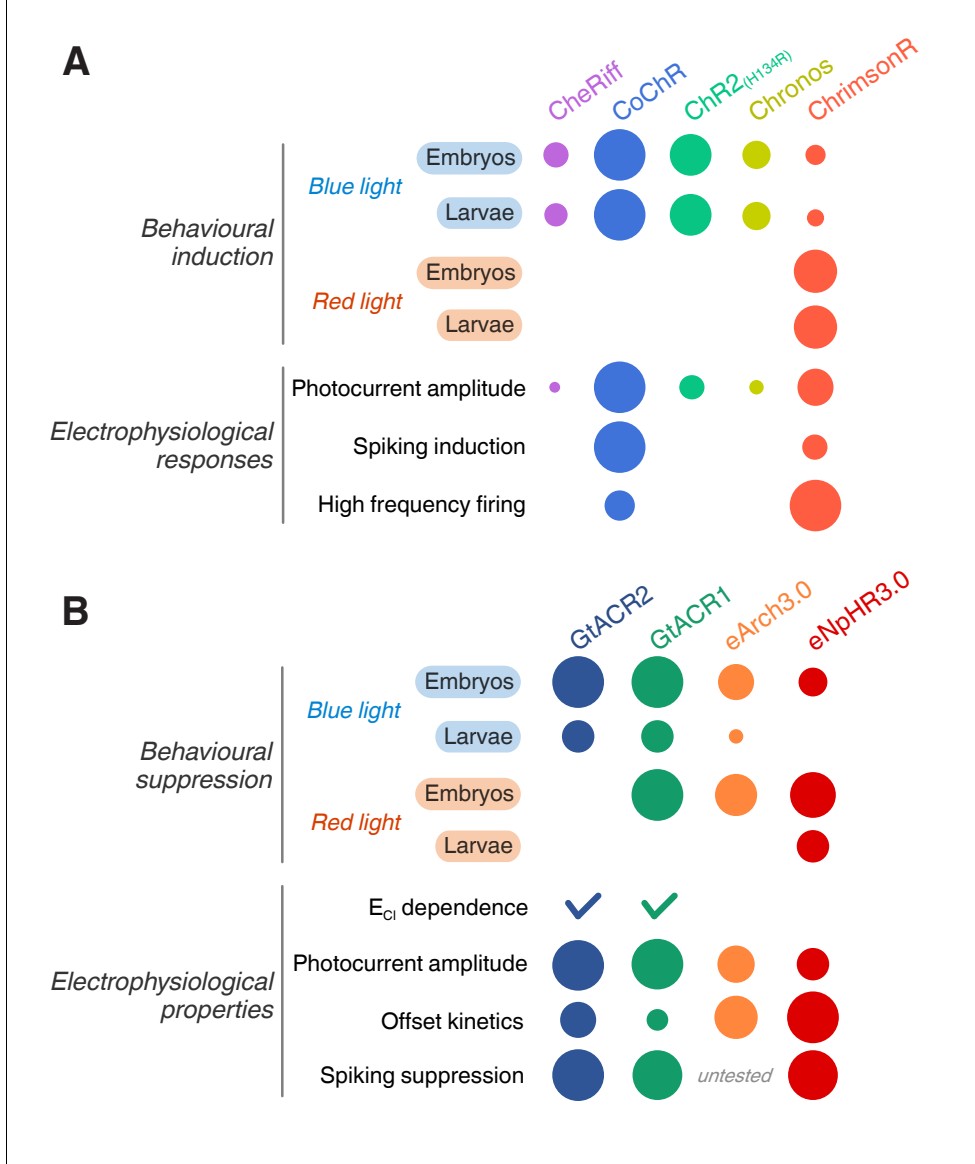

**Figure 10.** Summary of opsin line efficacy. (**A**) Efficacy of cation channelrhodopsin lines in inducing neural activity across behavioural assays, electrophysiological recordings, developmental stages and wavelengths. The radius of each circle is proportional to efficacy. (**B**) Efficacy of anion channelrhodopsins and Cl⁻/H⁺ pumps in suppressing neural activity.

## An in vivo platform for opsin tool selection

The selection of optogenetic actuators should be based on their ability to reliably control neural activity in vivo. While previous efforts compared opsin efficacy using transient expression strategies (e.g. through viral or plasmid-mediated opsin gene delivery, see *Mattis et al., 2011* and Introduction), here we calibrated opsin effects in stable transgenic lines, which offer more reproducible expression across experiments and laboratories (*Kikuta and Kawakami, 2009*; *Yizhar et al., 2011*). Overall, there was good qualitative agreement between behavioural and electrophysiological results, with efficacy in behavioural assays (even with transient expression) largely predicting rank order in photocurrent amplitudes. This illustrates the utility of high-throughput behavioural assays for rapid evaluation and selection of expression constructs prior to more time-consuming generation and characterisation of stable lines and electrophysiological calibration. We observed broad variation in efficacy across lines, likely attributable to differences in both the intrinsic properties of the opsin as

well as variation in expression and membrane targeting. Membrane trafficking can also be influenced by the fluorescent protein fused to the actuator (*Arrenberg et al., 2009*). In our hands, we observed better expression with the tdTomato fusion reported here than with previous attempts using a tagRFP fusion protein. In the future, expression might be further improved through codon optimisation (*Horstick et al., 2015*), trafficking-enhancing sequences (*Gradinaru et al., 2010*; *Mattis et al., 2011*), alternative expression targeting systems (*Luo et al., 2008*; *Sjulson et al., 2016*) and optimisation of the fluorescent reporter protein.

Behavioural and electrophysiological readouts complemented one another and enriched the interpretation of our results. Electrophysiological recordings in a defined cell type allowed direct and comparative calibration of photocurrents. Although several opsin lines did not evoke action potentials in low-input-resistance pMNs, behavioural assays showed that all lines induced tail movements in larvae. This is likely due to recruitment of secondary motor neurons labelled by the *Tg (mnx1:GAL4)* transgene, which have higher input resistance (*Menelaou and McLean, 2012*). Behavioural assays at multiple ages revealed that anion channelrhodopsins can excite neurons in one dpf embryos which was corroborated by making whole-cell recordings using a patch solution reproducing the high intracellular chloride concentration observed in embryonic neurons (*Reynolds et al., 2008*; *Zhang et al., 2010*).

Overall, our platform enables efficient selection and calibration of optogenetic tools for in vivo neuroscience. It also enables opsin-specific optimisation of light delivery (i.e. wavelength, pulse duration, frequency and intensity). For example, we found that equivalent stimulation regimes produced different rates of spiking adaptation that impacted the ability to control high-frequency firing, depending on the specific ospin line in question.

## Robust and precise optogenetic induction of spiking

Which opsin lines are best suited for reliable neural activation? Photocurrent amplitude, measured in pMNs, was proportional to estimated opsin expression level (*Figure 1—figure supplement 1D*) and was predictive of the ability of opsin lines to induce behaviour via activation of distinct cell types at both larval and embryonic stages (CoChR > ChrimsonR > ChR2$_{(H134R)}$ > Chronos $\geq$ CheRiff). The CoChR and ChrimsonR lines showed the highest expression levels among cation channelrhodopsins and were the only lines capable of inducing action potentials in pMNs, consistent with their photocurrent amplitudes exceeding pMN rheobase. Notably, CoChR evoked spikes in all pMNs tested and triggered behaviour with maximal response probability in larvae at irradiance levels as low as 0.63 mW/mm$^2$.

Where precise control of a cell's firing pattern is desired, electrophysiological calibration is essential to tune stimulation parameters for a specific opsin/cell-type combination. Our data indicate that in primary motor neurons, light pulses can lead to bursts of spikes and substantial firing rate adaptation during high-frequency stimulation, likely a result of plateau potentials and inactivation of voltage-gated sodium channels. Thus, although the CoChR line produced large-amplitude photocurrents and was highly efficient and precise in evoking the first spike, in this particular cell type it was also prone to burst firing even for short (0.5 ms) light pulses, which compromised spiking entrainment with high-frequency stimulations. However, CoChR has been used to elicit single spikes in mouse pyramidal cells with 1 ms light pulses at frequencies up to 50 Hz (*Ronzitti et al., 2017*). A thorough calibration in the cell type of interest in vivo is therefore necessary for precise control of spike number and timing. Compared to CoChR, we observed that ChrimsonR, although less effective in inducing firing in primary motor neurons overall, led to less spike adaptation during stimulation and fewer bursts of spikes.

## Excitatory effects of anion channelrhodopsins

Anion channelrhodopsins induced movements at light onset in one dpf embryos as well as transient spiking in pMNs when using an intracellular solution that mimicked the high ECl (–50 mV) of immature neurons. This is consistent with GtACRs functioning as a light-gated chloride conductance (*Govorunova et al., 2015*). The transient nature of spiking and motor activity might be due to the initial large inward photocurrent depolarising neurons above spiking threshold. Transient induction of action potentials with GtACRs has also been observed in rat cortical pyramidal neurons in brain slices (*Malyshev et al., 2017*) as well as cultured hippocampal neurons (*Mahn et al., 2018*) and has

been attributed to antidromic spiking resulting from a positively shifted ECl in the axon (*Mahn et al., 2016*; *Mahn et al., 2018*). In light of this, the use of GtACRs in immature neurons or subcellular structures should be carefully calibrated and use of Cl$^-$/H$^+$ pumps may be preferable. The likely mechanism of silencing induced by activation of GtACRs is shunting as the large photocurrents are associated with a reduction in the input resistance of the cell. In addition, GtACRs bring the membrane potential close to ECl, which may – depending on the physiological values of ECl in vivo – also lead to depolarisation block.

## Precise optogenetic inhibition of neural activity

To accurately suppress action potentials, opsin tools must be carefully selected with consideration for developmental stage and ECl-dependent effects as well as photocurrent kinetics. GtACRs generated large photocurrents with fast activation kinetics, which can explain why GtACR1 was effective in inhibiting single action potentials with short light pulses in larval pMNs. Cl$^-$/H$^+$ pump photocurrents instead showed fast deactivation kinetics, which allowed eNpHR3.0-expressing neurons to rapidly resume spiking at light offset. Differences in photocurrent kinetics between opsin classes – that is channels vs. pumps – may thus differentially affect the temporal resolution of activity inhibition and recovery, respectively. The combined behavioural and electrophysiological approach can be extended in the future to optogenetic silencers based on K$^+$ channel activation, such as the recently introduced PAC-K (*Bernal Sierra et al., 2018*).

In conclusion, our calibrated optogenetic toolkit and associated methodology provide an in vivo platform for designing controlled optogenetic experiments and benchmarking novel opsins.

# Materials and methods

**Key resources table**

| Reagent type (species) or resource | Designation | Source or reference | Identifiers | Additional information |
|---|---|---|---|---|
| Genetic reagent (*Danio rerio*) | Tg(UAS:ChrimsonR-tdTomato)u328Tg | This study | ZFIN ID: ZDB-ALT-190226–2 | Available from EZRC |
| Genetic reagent (*Danio rerio*) | Tg(UAS:Chronos-tdTomato)u330Tg | This study | ZFIN ID: ZDB-ALT-190226–3 | Available from EZRC |
| Genetic reagent (*Danio rerio*) | Tg(UAS:CoChR-tdTomato)u332Tg | This study | ZFIN ID: ZDB-ALT-190226–4 | Available from EZRC |
| Genetic reagent (*Danio rerio*) | Tg(UAS:CheRiff-tdTomato)u334Tg | This study | ZFIN ID: ZDB-ALT-190226–5 | Available from EZRC |
| Genetic reagent (*Danio rerio*) | Tg(UAS:GtACR1-tdTomato)u336Tg | This study | ZFIN ID: ZDB-ALT-190226–6 | Available from EZRC |
| Genetic reagent (*Danio rerio*) | Tg(UAS:GtACR2-tdTomato)u338Tg | This study | ZFIN ID: ZDB-ALT-190226–7 | Available from EZRC |
| Genetic reagent (*Danio rerio*) | Tg(UAS:eArch3.0-eYFP)mpn120 | This study | transgene | Available from Baier lab |
| Genetic reagent (*Danio rerio*) | Tg(UAS:eNpHR3.0-eYFP)mpn121 | This study | transgene | Available from Baier Lab |
| Genetic reagent (*Danio rerio*) | Tg(UAS:Cr.ChR2-YFP)icm11Tg | PMID:26752076 | ZFIN ID: ZDB-ALT-150324–2 | Available from EZRC (*Fidelin et al., 2015*) |
| Genetic reagent (*Danio rerio*) | Tg(UAS:GFP)zf82 | PMID:19835787 | ZFIN ID: ZDB-ALT-080528–1 | *Asakawa et al., 2008* |
| Genetic reagent (*Danio rerio*) | Tg(isl2b.2:GAL4-VP16, myl7:EGFP)zc60Tg | PMID:20702722 | ZFIN ID: ZDB-ALT-101130–1 | *Ben Fredj et al., 2010* |
| Genetic reagent (*Danio rerio*) | Tg(isl2b:GAL4-VP16, myl7:TagRFP)zc65 | PMID:21905164 | ZFIN ID: ZDB-FISH-150901–13523 | *Fujimoto et al., 2011* |
| Genetic reagent (*Danio rerio*) | Et(–0.6hsp70l:GAL4-VP16)s1020tEt | PMID:17369834 | ZFIN ID: ZDB-ALT-070420–21 | *Scott et al., 2007* |
| Genetic reagent (*Danio rerio*) | Tg(mnx1:GAL4) icm23Tg | PMID:26946992 | ZFIN ID: ZDB-ALT-160120–1 | *Böhm et al., 2016* |

*Continued on next page*

*Continued*

| Reagent type (species) or resource | Designation | Source or reference | Identifiers | Additional information |
|---|---|---|---|---|
| Genetic reagent (*Danio rerio*) | *Et(−109Xla.Eef1a1: GFP)mn2Et* | PMID:15347431 | ZFIN ID: ZDB-ALT-080625–1 | *Balciunas et al., 2004* |
| Recombinant DNA reagent | *pTol1-UAS:ChrimsonR-tdTomato* | This study | Addgene ID: 124231 | Available from Addgene |
| Recombinant DNA reagent | *pTol1-UAS:Chronos-tdTomato* | This study | Addgene ID: 124232 | Available from Addgene |
| Recombinant DNA reagent | *pTol1-UAS:CoChR-tdTomato* | This study | Addgene ID: 124233 | Available from Addgene |
| Recombinant DNA reagent | *pTol1-UAS:CheRiff-tdTomato* | This study | Addgene ID: 124234 | Available from Addgene |
| Recombinant DNA reagent | *pTol1-UAS:GtACR1-tdTomato* | This study | Addgene ID: 124235 | Available from Addgene |
| Recombinant DNA reagent | *pTol1-UAS:GtACR2-tdTomato* | This study | Addgene ID: 124236 | Available from Addgene |
| Recombinant DNA reagent | *pTol1-UAS:ChR2 (H134R)-tdTomato* | This study | Addgene ID: 124237 | Available from Addgene |
| Recombinant DNA reagent | *pTol2-UAS: eArch3.0-eYFP* | This study | plasmid | Available from Baier lab |
| Recombinant DNA reagent | *pTol2-UAS:eNpHR3.0-eYFP* | This study | plasmid | Available from Baier lab |
| Software, algorithm | MATLAB | MathWorks | RRID:SCR_001622 | https://uk.mathworks.com/products/matlab.html |
| Software, algorithm | Python | Anaconda | RRID:SCR_008394 | https://www.anaconda.com |
| Software, algorithm | LabView | National Instruments | RRID:SCR_014325 | http://www.ni.com/en-gb/shop/labview.html |
| Software, algorithm | Prism | GraphPad | RRID:SCR_002798 | https://www.graphpad.com/scientific-software/prism/ |

## Experimental model

Animals were reared on a 14/10 hr light/dark cycle at 28.5˚C. For all experiments, we used zebrafish (*Danio rerio*) embryos and larvae homozygous for the *mitfa^{w2}* skin-pigmentation mutation (*Lister et al., 1999*). All larvae used for behavioural assays were fed *Paramecia* from 4 dpf onward. Animal handling and experimental procedures were approved by the UCL Animal Welfare Ethical Review Body and the UK Home Office under the Animal (Scientific Procedures) Act 1986.

In vivo electrophysiological recordings were performed in 5–6 dpf zebrafish larvae from AB and Tüpfel long fin (TL) strains in accordance with the European Communities Council Directive (2010/63/EU) and French law (87/848) and approved by the Institut du Cerveau et de la Moelle épinière, the French ministry of Research and the Darwin Ethics Committee (APAFIS protocol #16469–2018071217081175 v5).

## Cloning and transgenesis

To generate the *UAS:opsin-tdTomato* DNA constructs used for transient opsin expression and for creating the stable *Tg(UAS:opsin-tdTomato)* transgenic lines, the coding sequences of the opsins listed below and the red fluorescent protein tdTomato (from *pAAV-Syn-Chronos-tdTomato*) were cloned in frame into a UAS Tol1 backbone (*pT1UciMP*).

The source plasmids used for cloning *UAS:opsin-tdTomato* DNA constructs were:

- ChrimsonR from *pCAG-ChrimsonR-tdT* (Addgene plasmid # 59169)
- Chronos from *pAAV-Syn-Chronos-tdTomato* (Addgene plasmid # 62726)
- CoChR from *pAAV-Syn-CoChR-GFP* (Addgene plasmid # 59070)
- CheRiff from *FCK-CheRiff-eGFP* (Addgene plasmid # 51693)
- GtACR1 from *pFUGW-hGtACR1-EYFP* (Addgene plasmid # 67795)

- GtACR2 from *pFUGW-hGtACR2-EYFP* (Addgene plasmid # 67877)
- ChR2$_{(H134R)}$ from *pAAV-Syn-ChR2(H134R)-GFP* (Addgene plasmid # 58880)

The *pCAG-ChrimsonR-tdT*, *pAAV-Syn-Chronos-tdTomato*, *pAAV-Syn-CoChR-GFP* and *pAAV-Syn-ChR2(H134R)-GFP* plasmids were gifts from Edward Boyden (*Boyden et al., 2005*; *Klapoetke et al., 2014*). The *FCK-CheRiff-eGFP* plasmid was a gift from Adam Cohen (*Hochbaum et al., 2014*). The *pFUGW-hGtACR1-EYFP* and *pFUGW-hGtACR2-EYFP* plasmids were gifts from John Spudich (*Govorunova et al., 2015*). The *pT1UciMP* plasmid was a gift from Harold Burgess (Addgene plasmid # 62215) (*Horstick et al., 2015*).

The cloning was achieved using the In-Fusion HD Cloning Plus CE kit (Clontech) with the following primers:

- ChrimsonR_fw, CTCAGCGTAAAGCCACCATGGGCGGAGCT
- Chronos_fw, CGTAAAGCCACCATGGAAACAGCC
- CoChR_fw, CTCAGCGTAAAGCCACCATGCTGGGAAACG
- CoChR_rev, TACTACCGGTGCCGCCACTGT
- CoChR_tdT_fw, ACAGTGGCGGCACCGGTAGTA
- CheRiff_fw, CTCAGCGTAAAGCCACCATGGGCGGAGCT
- CheRiff_rev, CTACCGGTGCCGCCACTTTATCTTCCTCTGTCACG
- CheRiff_tdT_fw, TAAAGTGGCGGCACCGGTAGTAGCAGTGAG
- GtACR1_fw, CTCAGCGTAAAGCCACCATGAGCAGCATCACCTGTGATC
- GtACR1_rev, CTACCGGTGCCGCGGTCTCGCCGGCTCTGG
- GtACR1_tdT_fw, CGAGACCGCGGCACCGGTAGTAGCAGTGAG
- GtACR2_fw, CTCAGCGTAAAGCCACCATGGCCTCCCAGGTCGT
- GtACR2_rev, CTACCGGTGCCGCCCTGCCGAACATTCTG
- GtACR2_tdT_fw, CGGCAGGGCGGCACCGGTAGTAGCAGTGAG
- ChR2(H134R)_fw, CTCAGCGTAAAGCCACCATGGACTATGGCGGCG
- ChR2(H134R)_rev, TACTCACTGCTACTACCGGTGCCGCCAC
- ChR2(H134R)_tdT_fw, ACCGGTAGTAGCAGTGAGTAAGG
- tdT_rev_40 bp, CTCGAGATCTCCATGTTTACTTATACAGCTCATCCATGCC
- tdT_rev_45 bp, CTAGTCTCGAGATCTCCATGTTTACTTATACAGCTCATCCATGCC

To generate the stable *Tg(UAS:opsin-tdTomato)* lines, purified *UAS:opsin-tdTomato* DNA constructs were first sequenced to confirm gene insertion and integrity and, subsequently, co-injected (35 ng/μl) with Tol1 transposase mRNA (80 ng/μl) into *Tg(KalTA4u508)* zebrafish embryos (*Antinucci et al., 2019*) at the early one-cell stage. Transient expression, visible as tdTomato fluorescence, was used to select injected embryos that were then raised to adulthood. Zebrafish codon-optimised *Tol1* transposase mRNA was prepared by in vitro transcription from NotI-linearised *pCS2-Tol1.zf1* plasmid using the SP6 transcription mMessage mMachine kit (Life Technologies). The *pCS2-Tol1.zf1* was a gift from Harold Burgess (Addgene plasmid # 61388) (*Horstick et al., 2015*). RNA was purified using the RNeasy MinElute Cleanup kit (Qiagen). Germ line transmission was identified by mating sexually mature adult fish to *mitfa^{w2/w2}* fish and subsequently examining their progeny for tdTomato fluorescence. Positive embryos from a single fish were then raised to adulthood. Once this second generation of fish reached adulthood, positive embryos from a single 'founder' fish were again selected and raised to adulthood to establish stable *Tg(KalTA4u508;UAS:opsin-tdTomato)* double-transgenic lines.

To generate the *UAS:opsin-eYFP* DNA constructs used for creating the stable *Tg(UAS:opsin-eYFP)* transgenic lines, the coding sequences of the opsins fused with eYFP listed below were cloned into a UAS Tol2 backbone (*pTol2 14xUAS:MCS*).

- *eArch3.0-eYFP* from *pAAV-CaMKIIa-eArch_3.0-EYFP* (Addgene plasmid # 35516)
- *eNpHR3.0-eYFP* from *pAAV-Ef1a-DIO-eNpHR 3.0-EYFP* (Addgene plasmid # 26966)

The *pAAV-CaMKIIa-eArch_3.0-EYFP* and *pAAV-Ef1a-DIO-eNpHR 3.0-EYFP* plasmids were gifts from Karl Deisseroth (*Gradinaru et al., 2010*; *Mattis et al., 2011*).

The coding sequences were amplified by PCR using the following primers and cloned into either EcoRI/NcoI (for eArch3.0) or EcoRI/SphI (for eNpHR3.0) sites of the *pTol2 14xUAS:MCS* plasmid:

- eArch3.0_fw, ATGAATTCGCCACCATGGACCCCATCGCTCT
- eArch3.0_rev, ATGCATGCTCATTACACCTCGTTCTCGTAG
- eNpHR3.0_fw, ATGAATTCGCCACCATGACAGAGACCCTGC

- eNpHR3.0_rev, TACCATGGTTACACCTCGTTCTCGTAGC

To generate the stable *Tg(UAS:opsin-eYFP)* lines, purified *UAS:opsin-eYFP* DNA constructs were first sequenced to confirm gene insertion and integrity and, subsequently, co-injected (25 ng/µl) with Tol2 transposase mRNA (25 ng/µl) into *Tg(isl2b:GAL4-VP16, myl7:TagRFP)zc65* (*Fujimoto et al., 2011*) (for eArch3.0-eYFP) or *Tg(s1020t:GAL4)* (*Scott et al., 2007*) (for eNpHR3.0-eYFP) zebrafish embryos at the early one-cell stage. Transient expression, visible as eYFP fluorescence, was used to select injected embryos that were then raised to adulthood. Zebrafish codon-optimised *Tol2* transposase mRNA was prepared by in vitro transcription from NotI-linearised *pCS2-zT2TP* plasmid using the SP6 transcription mMessage mMachine kit (Life Technologies). The *pCS2-zT2TP* was a gift from Koichi Kawakami (*Suster et al., 2011*). RNA was purified using the NucleoSpin Gel and PCR Clean-up kit (Macherey-Nagel). Germ line transmission was identified by mating sexually mature adult fish to *mitfa^{w2/w2}* fish and, subsequently, examining their progeny for eYFP fluorescence. Positive embryos from each injected fish were then raised to adulthood. Once this second generation of fish reached adulthood, positive embryos from a single 'founder' fish were again selected and raised to adulthood to establish stable *Tg(Isl2b:GAL4;UAS:eArch3.0-eYFP)* or *Tg(s1020t:GAL4;UAS:eNpHR3.0-eYFP)* double-transgenic lines.

## Fluorescence image acquisition

Zebrafish embryos or larvae were mounted in 1% low-melting point agarose (Sigma-Aldrich) and anesthetised using tricaine (MS-222, Sigma-Aldrich). Imaging was performed using a custom-built 2-photon microscope (XLUMPLFLN 20 × 1.0 NA objective [Olympus], 580 nm PMT dichroic, band-pass filters: 510/84 [green], 641/75 [red] [Semrock], R10699 PMT [Hammamatsu Photonics], Chameleon II ultrafast laser [Coherent Inc]). Imaging was performed at 1040 nm for opsin-tdTomato lines, while 920 nm excitation was used for opsin-eYFP lines. In both cases, the same laser power at sample (10.7 mW) and PMT gain were used. For the images displayed in *Figures 1C*, *3B* and *7B* and *Figure 7—figure supplement 3B*, equivalent imaging field of view and pixel size were used (1200 × 800 px, 0.385 µm/px). The imaging field of view and pixel size for images displayed in *Figures 2C* and *6B* were 960 × 680 px, 0.385 µm/px. For all these images, the same acquisition averaging (mean image from 12 frames) and z-spacing of imaging planes (2 µm) were used.

The image displayed in *Figure 4A* was acquired from a single plane on a fluorescence microscope (AxioExaminer D1 [Zeiss], 63 × 1.0 NA objective [Zeiss], Xcite [Xcelitas, XT600] 480 nm LED illumination, 38HE filtercube [Zeiss], ImagEM camera [Hammamatsu]), with an imaging field of view of 512 × 512 px and 0.135 µm/px pixel size.

## Opsin expression analysis

Image stacks were acquired from the spinal cord of 5 dpf *Tg(mnx1:GAL4;UAS:opsin-FP)* larvae using a 2-photon microscope and acquisition parameters described above. Maximum intensity z-projections spanning 5–10 µm in depth were used to estimate opsin expression at the plasma membrane of motor neurons. First, automated cell body segmentation was performed using Cellpose to obtain 'cell body masks' (*Stringer et al., 2020*; https://github.com/MouseLand/cellpose). Then, 'membrane masks' corresponding to outlines of the 'cell body masks' (see *Figure 1—figure supplement 1A*) were generated by running a boundary tracing routine for binary objects in MATLAB (MathWorks). For each cell, we computed the mean fluorescence intensity across all pixels in the corresponding membrane mask. Cells were grouped into primary or secondary motor neurons according to both area of cell body mask and location along the dorsal-ventral axis of the spinal cord (*Menelaou and McLean, 2012*). Cells with soma area larger than 60 µm² located in the dorsal half of the spinal cord were classified as primary motor neurons, cells with area smaller than 50 µm² were classified as secondary motor neurons (see *Figure 1—figure supplement 1B*).

## Behavioural assays

The same monitoring system was used for all behavioural assays (see schematic in *Figure 2A*) with some differences. Images were acquired under infrared illumination (850 nm) using a high-speed camera (Mikrotron MC1362, 500 µs shutter-time) equipped with a machine vision lens (Fujinon HF35SA-1) and an 850 nm bandpass filter to block visible light. The 850 nm bandpass filter was removed during embryonic activation assays (in which images were acquired at 1000 fps) to

determine time of light stimulus onset. In all other assays, lower acquisition rates were used (i.e. 50 or 500 fps) and, within each assay, the frames corresponding to stimulus onset/offset were consistent across trials.

Light was delivered across the whole arena from above using the following LEDs (spectral bandwidth at half maximum for each LED is reported in parenthesis):

*For embryonic assays*

- 470 nm OSRAM Golden Dragon Plus LED (LB W5AM; 25 nm).
- 590 nm ProLight LED (PM2B-3LAE-SD; 18 nm).

*For larval assays*

- 459 nm OSRAM OSTAR Projection Power LED (LE B P2W; 27 nm).
- 617 nm OSRAM OSTAR Projection Power LED (LE A P2W; 18 nm).

The 459 and 617 nm LEDs were projected onto the arena with an aspheric condenser with diffuser surface. Irradiance was varied using constant current drive electronics with pulse-width modulation at 5 kHz. Irradiance was calibrated using a photodiode power sensor (Thorlabs S121C). LED and camera control were implemented using LabVIEW (National Instruments).

Before experiments, animals were screened for opsin expression in the target neural population at either 22 hpf (embryonic assays) or 3 dpf (larval assays) using a fluorescence stereomicroscope (Olympus MVX10). For each opsin, animals with similar expression level were selected for experiments together with control opsin-negative siblings. To reduce variability in opsin expression level, all animals used for behavioural experiments were heterozygous for both the GAL4 and UAS transgenes. Animals were placed in the arena in the dark for around 2 min before starting experiments. For all assays, each light stimulus was repeated at least three times. Each trial lasted 1 s in behavioural activation assays and 30 s in behavioural inhibition assays.

## Embryonic activation assay

Opsin expression was targeted to trigeminal ganglion neurons using the *Tg(isl2b:GAL4)* transgene (*Ben Fredj et al., 2010*). Behaviour was monitored at 1000 fps across embryos (28–30 hpf) individually positioned in agarose wells (~2 mm diameter) in fish facility water and free to move within their chorion. Embryos were subjected to 5 or 40 ms pulses of blue (470 nm) or amber (590 nm) light at different irradiance levels (4.5–445 $\mu$W/mm$^2$) and with a 15 s inter-stimulus interval in the dark.

## Embryonic inhibition assay

Opsin expression was targeted to spinal primary and secondary motor neurons and interneurons (Kolmer-Agduhr cells and ventral longitudinal descending interneurons) using the *Tg(s1020t:GAL4)* transgene (*Scott et al., 2007*). Behaviour was monitored at 50 fps across embryos (24–27 hpf) individually positioned in agarose wells (~2 mm diameter) with fish facility water and free to move within their chorion. Embryos were subjected to 10 s pulses of blue (470 nm) or amber (590 nm) light at different irradiance levels (0–227 $\mu$W/mm$^2$) with a 50 s inter-stimulus interval in the dark.

## Larval activation assay

Opsin expression was targeted to primary and secondary spinal motor neurons using the *Tg(mnx1: GAL4)* transgene (*Böhm et al., 2016*). Behaviour was monitored at 500 fps in 6 dpf larvae with their head restrained in 2% low-melting point agarose (Sigma-Aldrich) and their tail free to move. Larvae were subjected to 2 or 10 ms pulses of blue (459 nm) or red (617 nm) light at different irradiance levels (0.04–2.55 mW/mm$^2$) with a 20 s inter-stimulus interval in the dark. We also provided 250 ms trains of light pulses (1 ms pulse duration for blue light at 2.55 mW/mm$^2$ or 10 ms for red light at 1 mW/mm$^2$) at two pulse frequencies (20 or 40 Hz).

## Larval inhibition assays

Opsin expression was targeted to spinal cord neurons using either the *Tg(s1020t:GAL4)* or *Tg(mnx1: GAL4)* transgene, as above. Behaviour was monitored at 50 fps across 6 dpf larvae individually positioned in agarose wells (~1.4 cm diameter) with fish facility water in which they were free to swim. Larvae were subjected to 10 s pulses of blue (459 nm) or red (617 nm) light at different irradiance

levels (0.24–2.55 mW/mm$^2$) with a 50 s inter-stimulus interval in the dark. Control trials during which no light pulse was provided were interleaved between light stimulation trials.

## Behavioural data analysis

Movie data was analysed using MATLAB (MathWorks). Region of interests (ROIs) containing individual fish were manually specified. For each ROI, the frame-by-frame change in pixel intensity – ΔPixel – was computed in the following way. For each trial, pixel intensity values were low-pass filtered across time frames and the absolute frame-by-frame difference in intensity (*dI*) was obtained for each pixel. Pixels showing the highest variance in *dI* (top 5$^{th}$ percentile) were selected to compute their mean *dI,* corresponding to the ROI ΔPixel trace for the trial.

With the exception of the larval inhibition assay (see below), onset and offset of animal movements were detected from ΔPixel traces in the following way. For each ROI, ΔPixel traces were concatenated across all trials to estimate the probability density function (*pdf*) of ΔPixel values. The portion of the distribution with values below the *pdf* peak was mirror-reflected about the *x*-axis and a Gaussian was fitted to the obtained symmetric distribution. The mean ($\mu$) and standard deviation (σ) of the fitted Gaussian were then used to compute ROI-specific ΔPixel thresholds for detecting onset ($\mu + 6\sigma$) and offset ($\mu + 3\sigma$) of animal movements.

For embryonic and larval activation assays, behavioural response latency corresponds to the time from light stimulus onset to the start of the first detected movement. Movements were classified as optogenetically-evoked if their response latency was shorter than 200 ms for the embryonic assay or 50 ms for the larval assay, which corresponds to the minimum in the *pdf* of response latency from all opsin-expressing larvae (*Figure 3E*). For each animal, response probability to each light stimulus type corresponds to the fraction of trials in which at least one optogenetically-evoked movement was detected.

In the larval activation assay, the tail was tracked by performing consecutive annular line-scans, starting from a manually-selected body centroid and progressing towards the tip of the tail so as to define nine equidistant x-y coordinates along the tail. Inter-segment angles were computed between the eight resulting segments. Reported tail curvature was computed as the sum of these inter-segment angles. Rightward bending of the tail is represented by positive angles and leftward bending by negative angles. Number of tail beats corresponds to the number of full tail oscillation cycles. Tail theta-1 angle is the amplitude of the first half beat. Tail beat frequency was computed as the reciprocal of the mean full-cycle period during the first four tail oscillation cycles of a swim bout. Bout duration was determined from ΔPixel traces using the movement onset/offset thresholds described above.

For larval inhibition assays, images were background-subtracted using a background model generated over each trial (30 s duration). Images were then thresholded and the fish body centroid was found by running a particle detection routine for binary objects within suitable area limits. Tracking of body centroid position was used to compute fish speed, and periods in which speed was higher than 1 mm/s were classified as swim bouts. Bout speed was computed as the mean speed over the duration of each bout.

To account for group differences in baseline coil/bout rate and bout speed in inhibition assays, data was normalised at a given irradiance level by dividing by the mean rate/speed across fish in control (no light) trials.

## Electrophysiological recordings

### Transgenic lines

Opsin expression was targeted to primary motor neurons using the *Tg(mnx1:GAL4)* transgene (*Böhm et al., 2016*) with one exception: 11 out of 19 eNpHR3.0-expressing cells were recorded in *Tg(s1020t:GAL4)* larvae (*Scott et al., 2007*). As in behavioural assays, all animals used for electrophysiological experiments were heterozygous for both the GAL4 and UAS transgenes. For control recordings, we targeted opsin-negative GFP-expressing primary motor neurons in *Tg(mnx1:GAL4; UAS:EGFP)* (*Asakawa et al., 2008*) or *Tg(parga-GFP)* (*Balciunas et al., 2004*) larvae. In all transgenic lines used, primary motor neurons could be unambiguously identified as the 3–4 largest cell somas, located in the dorsal-most portion of the motor column (*Beattie et al., 1997*; *Bello-Rojas et al., 2019*). We verified primary motor neuron identity in a small subset of recordings from eYFP-

expressing cells in *Tg(mnx1:GAL4;UAS:ChR2(H134R)-eYFP)* larvae by adding 0.025% sulforhodamine-B acid chloride dye in the intracellular solution (Sigma-Aldrich) and filling the neuron to reveal its morphology. To maximise data acquisition in our in vivo preparation, when the first attempts of primary motor neuron recordings were not successful, we recorded neighbouring, dorsally-located presumed secondary motor neurons (11 out of 90 included cells).

## Data acquisition

Zebrafish larvae (5–6 dpf) were first paralysed in 1 mM α-Bungarotoxin solution (Tocris) for 3–6 min after which they were pinned in a lateral position to a Sylgard-coated recording dish (Sylgard 184, Dow Corning) with tungsten pins inserted through the notochord. The skin was removed between the trunk and midbody regions using sharp forceps, after which the dorsal muscle from 2 to 3 somites was suctioned with glass pipettes (~50 μm opening made from capillaries of 1.5 mm outer diameter, 1.1 mm inner diameter; Sutter). Patch pipettes were made from capillary glass (1 mm outer diameter, 0.58 mm inner diameter; WPI) with a horizontal puller (Sutter Instrument P1000) and had resistances between 8–16 MΩ. To first pass the dura, we applied a higher positive pressure (30–40 mm Hg) to the recording electrode via a pneumatic transducer (Fluke Biomedical, DPM1B), which was then lowered (20–25 mm Hg) once the electrode was near the cells. We generally recorded data from a single cell per larva. In a few instances, two cells from separate adjacent somites were recorded in the same fish.

External bath recording solution contained the following: 134 mM NaCl, 2.9 mM KCl, 2.1 mM $CaCl_2-H_2O$, 1.2 mM $MgCl_2$, 10 mM glucose, and 10 mM HEPES, with pH adjusted to 7.8 with 9 mM NaOH and an osmolarity of 295 mOsm. We blocked glutamatergic and GABAergic synaptic transmission with a cocktail of: 20 μM CNQX or DNQX, 50 μM D-AP5, 10 μM Gabazine (Tocris) added to the external recording solution. The –50 mV ECl solution contained: 115 mM K-gluconate, 15 mM KCl, 2 mM $MgCl_2$, 4 mM Mg-ATP, 0.5 mM EGTA, 10 mM HEPES, with pH adjusted to 7.2 with 11 mM KOH solution, and a 285 mOsm. In these conditions, we calculated the liquid junction potential (LJP; Clampfit calculator) to be 12.4 mV. The –70 mV ECl solution contained: 126 mM K-gluconate, 4 mM KCl, 2 mM $MgCl_2$, 4 mM Mg-ATP, 0.5 mM EGTA, 10 mM HEPES, pH adjusted to 7.2 with 11 mM KOH solution, 285 mOsm and a 13.3 mV LJP. All reagents were obtained from Sigma-Aldrich unless otherwise stated.

Recordings were made with an Axopatch 700B amplifier and digitised with Digidata 1440A or 1550B (Molecular Devices). pClamp software was used to acquire electrophysiological data at a sampling rate of 20 kHz and low-pass filtered at 2 kHz (voltage clamp) or 10 kHz (current clamp). Voltage clamp recordings were acquired with full whole-cell compensation and ~60% series resistance compensation, while corrections for bridge balance and electrode capacitance were applied in current clamp mode. Cells were visualised with a 63×/1.0 NA or a 60×/1.0 NA water-immersion objective (Zeiss or Nikon, respectively) on a fluorescence microscope equipped with differential interference contrast optics (AxioExaminer D1, Zeiss or Eclipse FN1, Nikon).

## Optogenetic stimulation

Light stimulation was performed with either a X-Cite (Xcelitas, XT600) or a broadband white LED (Prizmatix, UHP-T-HCRI_DI) light source equipped with a combination of different bandpass and neutral density filters to modulate irradiance at specific wavelengths (see *Figure 4—figure supplement 1A* and *Supplementary file 4* for centre wavelengths/bandwidth and irradiance levels used to activate opsins). The onset, duration and irradiance level of light pulses were triggered and controlled via the Digidata device used for electrophysiological recordings.

For all cells, data was acquired in the following order: (1) series resistance was checked at the beginning, middle and end of recording; (2) action potential rheobase was determined by injecting 5 ms pulses of current (160–340 pA) in current-clamp gap-free mode; (3) voltage clamp recording of opsin photocurrents; (4) current clamp recording of voltage responses induced by opsin activation. Light stimuli were provided from low to high irradiance levels across all protocols. For each protocol, inter-stimulus intervals were between 10 and 15 s.

For cation channelrhodopsins, we used a range of short light pulses. Voltage clamp recordings were paired with a 5 ms light pulse, while current clamp recordings were performed with 0.1, 0.5, 1, 2 or 5 ms pulses. In addition, we tested whether we could optogenetically entrain neurons to spike

at frequencies ranging from 1 to 100 Hz using stimulus trains composed of 0.5, 1, 2 or 5 ms light pulses.

For anion channelrhodopsins and $Cl^-/H^+$ pumps, voltage and current clamp recordings were paired with a 1 s light pulse. In addition, we used two different tests of optogenetic inhibition during active spiking. To assess single spike inhibition efficacy and precision, we evoked spiking by injecting 5 ms pulses of current at 1.2–1.5 × rheobase for 10 trains at 5 Hz (1 s inter-train interval, total of 100 spikes triggered in 30 s), during which we provided 5 ms light pulses paired to the first current stimulus of the train and a subsequent one with progressively longer latency (*Zhang et al., 2007*). To test opsin ability to inhibit tonic firing over longer time periods, we evoked spiking with longer pulses of current (200–800 ms) at 1.2–1.5 × rheobase paired with a light pulse (50–200 ms) in the middle of the current stimulation. We first recorded a control current injection-only trial, followed by current and light pulse trials with a 20 s inter-stimulus interval.

## Data analysis

Data were analysed using the pyABF module in Spyder (3.3.6 MIT, running Python 3.6, scripts available here: https://github.com/wyartlab/Antinucci_Dumitrescu_et_al_2020; *Dumitrescu, 2020*; copy archived at https://github.com/elifesciences-publications/Antinucci_Dumitrescu_et_al_2020), MAT-LAB (MathWorks) and Clampfit (Molecular Devices). Series resistance (Rs) was calculated as a cell response to a 5 or 10 mV hyperpolarisation step in voltage clamp from a holding potential of –60 mV, with whole-cell compensation disabled. Membrane resistance (Rm) was obtained from the steady holding current at the new step, and membrane capacitance (Cm) corresponds to the area under the exponentially decaying current from peak to holding. We used the following cell inclusion criteria: (1) cell spiking upon injection of a 5 ms pulse of current; (2) membrane resting potential < –50 mV at all times; (3) > 150 pA current injection necessary to maintain the cell at a holding potential equal to resting potential in current clamp; (4) series resistance < 6 × pipette resistance at all times during the recording. We chose this conservative series resistance range as per previous electrophysiological procedures in other animal models, i.e. mammalian in vivo recordings with pipette resistance between 4–7 MΩ and max series resistance between 10–100 MΩ (*Margrie et al., 2002*). All reported membrane voltages were liquid junction potential corrected.

For voltage clamp recordings, we measured the maximum photocurrent amplitude in a time window of 100 ms (for cation channelrhodopsins) or 1 s (for anion channelrhodopsins and $Cl^-/H^+$ pumps) duration starting from light onset. To characterise photocurrent kinetics of cation channelrhodopsins, we measured the time to peak photocurrent from light onset (i.e. activation time) and computed the response decay time constant by fitting a monoexponential decay function to the photocurrent from peak to baseline (i.e. deactivation time constant). To compute photocurrent kinetics of anion channelrhodopsins and $Cl^-/H^+$ pumps, we fitted monoexponential functions to the following components of the response: activation time constant was computed from light onset to peak response, inactivation time constant from peak response to steady state (last 5 ms of light stimulation), deactivation time constant from steady state to baseline (1 s following light offset).

To characterise voltage responses induced by opsins under current clamp, we first classified events as spikes (when max voltage depolarisation was > –30 mV) or sub-threshold (peak voltage deflection < –30 mV). For each response type, we measured the absolute peak of the response, the time to reach maximum response from light onset and the time-decay to baseline from peak by fitting a monoexponential decay function, as above. To assess firing pattern fidelity, we calculated the number of spikes per light pulse in a train, the latency from light onset to the first spike occurring within a 10 ms time window, and the spike jitter as the standard deviation of spike latency values across a pulse train with given frequency.

Opsin efficacy in inhibiting single spikes was quantified using the following equation:

$$I = \frac{S_C - S_{C+L}}{S_C} \times 100$$

where $S_C$ is the mean number of spikes elicited by current pulses when no light was provided, $S_{C+L}$ is the mean number of spikes elicited during time periods in which a light pulse was paired with a current pulse, and $I$ is the inhibition index (100% being perfect inhibition and negative values indicating additional spikes were generated during light pulses). Tonic firing inhibition efficacy was

quantified by counting the number of spikes occurring during the light delivery period and normalising this count to provide spikes generated per 50 ms.

## Statistical analysis

All statistical analyses were performed using Prism (GraphPad). Sample distributions were first assessed for normality and homoscedasticity. Details regarding the statistical tests used are reported in *Supplementary file 2* for behavioural data and *Supplementary file 3* for electrophysiological data. Significance threshold was set to 0.05 and all reported p-values were corrected for multiple comparisons. Tests were two-tailed for all experiments. Statistical analysis performed during the peer-review process has been reported as exploratory analyses (see *Supplementary file 3*). Number of animals/cells are provided for each graph. No outliers were excluded from the analyses.

## Acknowledgements

The authors thank members of the Bianco lab and Wyart lab for helpful discussions. We thank staff from the UCL and ICM PHENO fish facilities for fish care and husbandry (UCL: Carole Wilson and team; ICM: Sophie Nunes-Figueiredo, Bogdan Buzurin, Monica Dicu and Antoine Arneau). Irene Arnold-Ammer and Enrico Kühn (MPI of Neurobiology) assisted with generation of transgenic lines. We also acknowledge support from the ICM electrophysiology core facility (CELIS-ePhys) and Mingyue Wu for electrophysiology experimental assistance. PA was supported by a Sir Henry Wellcome Postdoctoral Fellowship (204708/Z/16/Z). AD was supported by a Marie Curie Incoming International Fellowship (H2020-MSCA-IF-2016 Project #752199). FK was supported by a HFSP long-term fellowship (LT393/2010). Generation of opsins in HB's laboratory was supported by the Max Planck Society (HB and FK) and the DFG (SPP1926 Next-Generation Optogenetics). A Sir Henry Dale Fellowship from the Royal Society & Wellcome Trust (101195/Z/13/Z) and a UCL Excellence Fellowship were awarded to IHB. CW was funded by Human Frontier Science Program (HFSP) Research Grant (RGP063-2018) and the New York Stem Cell Foundation (NYSCF-R-NI39). The work on electrophysiological calibration of opsins performed in ICM has also received funding from the ICM foundation, the program 'Investissements d'avenir' ANR-10-IAIHU-06 (Big Brain Theory ICM Program), ANR-11-INBS-0011 (NeurATRIS: Translational Research Infrastructure for Biotherapies in Neurosciences).

## Additional information

### Competing interests

Claire Wyart: Reviewing editor, *eLife*. The other authors declare that no competing interests exist.

### Funding

| Funder | Grant reference number | Author |
| --- | --- | --- |
| Horizon 2020 Framework Programme | Marie Curie Incoming International Fellowship, H2020-MSCA-IF-2016 Project #752199 | Adna Dumitrescu |
| Human Frontier Science Program | RGP063-2018 | Claire Wyart |
| New York Stem Cell Foundation | NYSCF-R-NI39 | Claire Wyart |
| Wellcome | Sir Henry Wellcome Postdoctoral Fellowship, 204708/Z/16/Z | Paride Antinucci |
| Wellcome | Sir Henry Dale Fellowship, 101195/Z/13/Z | Isaac H Bianco |
| Royal Society | Sir Henry Dale Fellowship, 101195/Z/13/Z | Isaac H Bianco |
| University College London | Excellence Fellowship | Isaac H Bianco |

| Human Frontier Science Program | Long-term Fellowship, LT393/2010 | Fumi Kubo |
| --- | --- | --- |
| Deutsche Forschungsgemeinschaft | Next-Generation Optogenetics, SPP1926 | Herwig Baier |
| Max Planck Society | | Herwig Baier |
| Agence Nationale de la Recherche | ANR-10-IAIHU-06 | Charlotte Deleuze Claire Wyart |
| Agence Nationale de la Recherche | ANR-11-INBS-0011 | Charlotte Deleuze Claire Wyart |

The funders had no role in study design, data collection and interpretation, or the decision to submit the work for publication.

### Author contributions

Paride Antinucci, Adna Dumitrescu, Conceptualization, Resources, Data curation, Software, Formal analysis, Funding acquisition, Validation, Investigation, Visualization, Methodology, Writing - original draft, Writing - review and editing; Charlotte Deleuze, Conceptualization, Resources, Data curation, Methodology; Holly J Morley, Kristie Leung, Tom Hagley, Investigation; Fumi Kubo, Resources, Writing - review and editing; Herwig Baier, Resources, Supervision, Funding acquisition, Writing - review and editing; Isaac H Bianco, Conceptualization, Resources, Software, Supervision, Funding acquisition, Writing - original draft, Project administration, Writing - review and editing; Claire Wyart, Conceptualization, Resources, Software, Formal analysis, Supervision, Funding acquisition, Methodology, Writing - original draft, Project administration, Writing - review and editing

### Author ORCIDs

Paride Antinucci (iD) https://orcid.org/0000-0003-0573-5383
Adna Dumitrescu (iD) https://orcid.org/0000-0002-7354-1452
Holly J Morley (iD) http://orcid.org/0000-0002-0007-3563
Herwig Baier (iD) http://orcid.org/0000-0002-7268-0469
Isaac H Bianco (iD) https://orcid.org/0000-0002-3149-4862
Claire Wyart (iD) https://orcid.org/0000-0002-1668-4975

### Ethics

Animal experimentation: All larvae used for behavioural assays were fed Paramecia from 4 dpf onward. Animal handling and experimental procedures were approved by the UCL Animal Welfare Ethical Review Body and the UK Home Office under the Animal (Scientific Procedures) Act 1986. In vivo electrophysiological recordings were performed in 5-6 dpf zebrafish larvae from AB and Tüpfel long fin (TL) strains in accordance with the European Communities Council Directive (2010/63/EU) and French law (87/848) and approved by the Institut du Cerveau et de la Moelleépinière, the French ministry of Research and the Darwin Ethics Committee (APAFIS protocol #16469-2018071217081175v5).

### Decision letter and Author response

Decision letter https://doi.org/10.7554/eLife.54937.sa1
Author response https://doi.org/10.7554/eLife.54937.sa2

## Additional files

### Supplementary files

- Supplementary file 1. Summary table with properties of selected opsins.

- Supplementary file 2. Statistical tests used for behavioural data.

- Supplementary file 3. Statistical tests used for electrophysiological data.

- Supplementary file 4. Centre wavelength and bandwidth of light stimuli used in electrophysiological recordings.

• Transparent reporting form

## Data availability

All data generated or analysed during this study are included in the manuscript and supporting files. Source data files have been provided for all figures.

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
