## [Decision Letter]

**Acceptance summary:**

Engineered optogenetic actuators are critically important tools for modern circuit neuroscience studies but selection of the most experimentally suitable opsin has been challenging. By meticulously comparing properties and efficacy of activators and inhibitors using in vivo physiology and behavioral tests in transgenic zebrafish, this work provides clear use guidelines and a resource that will accelerate neural circuit studies.

**Decision letter after peer review:**

Thank you for submitting your article "A calibrated optogenetic toolbox of stable zebrafish opsin lines" for consideration by *eLife*. Your article has been reviewed by three peer reviewers, including Harold Burgess as the guest Reviewing Editor and Reviewer #1, and the evaluation has been overseen by Didier Stainier as the Senior Editor. The following individuals involved in review of your submission have agreed to reveal their identity: Michael Orger and Sabine Renninger (co-reviewers, Reviewer #2); and David L McLean (Reviewer #3).

The reviewers have discussed the reviews with one another and the Reviewing Editor has drafted this decision to help you prepare a revised submission.

The authors systematically characterized state-of-the-art optogenetic tools in zebrafish, testing experimental utility using behavioral tests and in vivo cellular physiology. The reviewers concurred that this is an exceptionally thorough study that exceeds previous work, and establishes benchmark criteria for assessing future reagents. This, and the suite of new tools provided, will have a large impact on neurobiological work in zebrafish. The results are compelling and could probably be published as is, however the reviewers suggested several points – not requiring any new experiments – that the authors should address to clarify the manuscript for the reader.

Essential revisions:

1) In Figure 2 and Figure 2—figure supplements 1 and 2, it is confusing that no responses are observed before the offset of the light pulse. If this is a limitation of the experimental conditions or image processing being used to detect movement, please explain as it makes it hard to judge the latency of responses compared to other assays in the paper and might result in an underestimation of responses. Data in other figures suggest that responses in <40 ms should be possible, so it would be helpful to clarify this point.

2) Although the authors mention (subsection “in vivo whole-cell recording of photocurrents in larval primary motor neurons”) that opsin-expression motor neurons have similar baseline physiology to opsin negative neurons, the two most effective opsins, CoChR and ChrimsonR appear to have properties that look different and in fact the statistics in Supplementary file 3 do indicate a significant effect of group for Figure 4B/C. Although the control v CoChR/Chrimson is not significant after multiple comparison adjustment, it is not clear that all 45 comparisons are relevant (and need to be corrected for) instead of just the 9 comparisons against control. Alerting readers to the possibility that expression may alter baseline membrane properties does not reduce the utility of these reagents – it emphasizes the importance of appropriate experimental controls.

3) Why does suppression of motor neuron excitability reduce the initiation frequency of swim bouts in Figure 7D ? If s1020 labels motor-neurons shouldn't there be a similar frequency of bouts, but with reduced bend amplitudes and speed of movement as for the mnx1 data in Figure 7—figure supplements 3/4?

4) Discussion: subsection “Excitatory effects of anion channelrhodopsins”, can you provide a reference for 'clamping at ECl' representing depolarization block? I was under the impression this occurred at more positive potentials. As described, this sounds more like 'shunting' by clamping at ECl as opposed to depolarization block.

---

## [Author Response]

Essential revisions:1) In Figure 2 and Figure 2—figure supplements 1 and 2, it is confusing that no responses are observed before the offset of the light pulse. If this is a limitation of the experimental conditions or image processing being used to detect movement, please explain as it makes it hard to judge the latency of responses compared to other assays in the paper and might result in an underestimation of responses. Data in other figures suggest that responses in <40 ms should be possible, so it would be helpful to clarify this point.

In the embryonic behavioural activation assay presented in Figure 2 and Figure 2—figure supplements 1 and 2, images were acquired at 1,000 fps with the 850 nm bandpass filter removed from the camera to accurately determine the time of light stimulus onset/offset. Consequently, we could not assess movement of the animals during the light pulse (when the image was saturated). This limitation will not result in underestimation of response probability for two reasons: (1) a movement initiated during the light pulse inevitably results in increased ΔPixel following stimulus offset, as the new position of the embryo produces an image different from the images before stimulus onset; (2) escape bouts last several tens to hundreds of ms (92 ± 65 ms, mean ± SD; see also Figure 2E and Video 1), hence embryos that initiated escape during the light stimulus would still move after stimulus offset. However, mean response latency for 40 ms pulses cannot be accurately determined as a fraction of responses are likely to initiate during the pulse (see truncated histograms in Figure 2—figure supplement 2). This limitation is very unlikely to apply to the shorter 5 ms pulse condition where response histograms indicate shortest latency responses begin well after stimulus offset. Therefore we only present mean latency estimates for this short pulse condition in Figure 2.

In all other assays, response initiation could always be detected because a 850 nm bandpass filter was used to block stimulation light.

*2) Although the authors mention (subsection “*in vivo *whole-cell recording of photocurrents in larval primary motor neurons”) that opsin-expression motor neurons have similar baseline physiology to opsin negative neurons, the two most effective opsins, CoChR and ChrimsonR appear to have properties that look different and in fact the statistics in* Supplementary file 3 *do indicate a significant effect of group for* Figure 4B/C*. Although the control v CoChR/Chrimson is not significant after multiple comparison adjustment, it is not clear that all 45 comparisons are relevant (and need to be corrected for) instead of just the 9 comparisons against control. Alerting readers to the possibility that expression may alter baseline membrane properties does not reduce the utility of these reagents – it emphasizes the importance of appropriate experimental controls.*

We a priori decided to perform all 45 pairwise post-hoc comparisons because the purpose of our experiment was to compare the efficiency of the opsins against each other, as well as with control opsin-negative neurons.

However, we agree that it is also desirable to correct for pairwise comparisons of the membrane properties for each opsin vs. control cells only. We performed the analysis suggested by the reviewer for Figure 4B, C and Figure 4—figure supplement 1C, D (see Exploratory Analysis in Supplementary file 2). The results are very similar and do not change the original interpretation of the data.

The only notable difference is that ChR2-YFP- and eNpHR3.0-YFP- positive neurons had slightly-depolarized resting membrane potentials (~+ 4-5mV) compared with control neurons. However, in the case of ChR2-YFP- and eNpHR3.0-YFP- positive neurons only, recordings were made on a setup on which cells displayed ~4 mV more depolarised resting membrane potentials compared to the setup used for control cells (see Panel A in Author response image 1). Therefore this minor effect could be due to a technical aspect of data collection, rather than an effect of opsin expression.

**Author response image 1. respfig1:** A Resting membrane potential plotted split by opsin type and recording rig (top) or grouped simply by experimenter (bottom). We found that cells recorded on Rig 2 had more depolarised resting membrane potentials (Rig 1 -77.4 ± 3mV N = 54, Rig 2 -73 ± 3.8mV N = 36; Mann-Whitney U = 310, p >0.0001). B Example responses from a CoChR and a GtACR1 neuron stimulated with a 1s long light pulse (bottom) and a zoomed in part of the trace (top shows boxed area from bottom).

We have kept the original multiple comparisons analysis in the revised manuscript, but also report this more restricted comparison in the Exploratory Analysis section from Supplementary file 2.

3) Why does suppression of motor neuron excitability reduce the initiation frequency of swim bouts in Figure 7D ? If s1020 labels motor-neurons shouldn't there be a similar frequency of bouts, but with reduced bend amplitudes and speed of movement as for the mnx1 data in Figure 7—figure supplements 3/4?

The Tg(s1020t:Gal4) line labels spinal interneurons as well as motor neurons (Wyart et al., 2009). Silencing spinal motor neurons and interneurons is expected to result in reduced recruitment probability of motor neurons in response to descending motor commands and therefore decreased bout initiation frequency, consistent with previous studies (e.g., Sternberg et al., 2016).

4) Discussion: subsection “Excitatory effects of anion channelrhodopsins”, can you provide a reference for 'clamping at ECl' representing depolarization block? I was under the impression this occurred at more positive potentials. As described, this sounds more like 'shunting' by clamping at ECl as opposed to depolarization block.

GtACR-expressing neurons showed transient spiking in response to light, followed by a rapid decrease in inward current but sustained depolarisation of the neuron. We think it is likely that silencing results from a combination of both shunting inhibition and depolarisation block. In support of the latter mechanism, it is known that cation channel rhodopsins induce depolarisation block (Herman et al., 2014) and indeed, in our recordings with 1s light pulses, CoChR-positive neurons spiked in the first ~ 50 ms after light onset and thereafter remained depolarised at ~60 mV for the remainder of the light stimulation (LPJ corrected, Author response image 1). GtACR1-positive neurons showed a similar response, except that membrane potential reached embryonic ECl (- 50 mV). Because this is more depolarised relative to the CoChR condition, we think depolarisation block likely contributes to neuronal silencing during prolonged GtACR stimulation. Overall, the transient spiking followed by silencing with membrane potential at ECl is therefore likely due to both depolarization block as well as shunting related to the large GtACR-evoked photocurrents. We added these interpretations to the revised manuscript.